# VER: Vision Expert Transformer for Robot Learning via Foundation Distillation and Dynamic Routing

**Yixiao Wang**[1], **Mingxiao Huo**[2], **Zhixuan Liang**[3], **Yushi Du**[4], **Lingfeng Sun**[1], **Haotian Lin**[2],
**Jinghuan Shang**[5], **Chensheng Peng**[1], **Mohit Bansal**[6], **Mingyu Ding**[6*], **Masayoshi Tomizuka**[1]
[1]UC Berkeley    [2]Carnegie Mellon University    [3]University of Hong Kong
[4]Peking University    [5]Stony Brook University    [6]UNC-Chapel Hill

## Abstract

Pretrained vision foundation models (VFMs) advance robotic learning via rich visual representations, yet individual VFMs typically excel only in specific domains, limiting generality across tasks. Distilling multiple VFMs into a unified representation can mitigate this limitation but often yields inflexible task-specific feature selection and requires costly full retraining to incorporate robot-domain knowledge. We propose **VER**, a **V**ision **E**xpert transformer for **R**obot learning. During pretraining, VER distills multiple VFMs into a vision expert library. We then finetune only a lightweight routing network (fewer than 0.4% of parameters) to dynamically select task-relevant experts from the pretrained library for downstream robot tasks. We further introduce *Patchwise Expert Routing* with *Curriculum Top-K Annealing* to improve both flexibility and precision of dynamic expert selection. Moreover, VER supports parameter-efficient finetuning for scalable expert utilization and adaptive robot-domain knowledge integration. Across 17 diverse robotic tasks and multiple policy heads, VER achieves state-of-the-art performance. We find that VER reduces large-norm outliers in task-irrelevant regions (e.g., background) and concentrates on task-critical regions. More visualizations and codes are available in `https://yixiaowang7.github.io/ver_page/`.

## 1 Introduction

Developing robotic systems capable of perceiving and interacting with complex, unstructured environments remains a fundamental challenge in embodied AI. Recently, visuomotor robot policy learning has emerged as a promising approach, enabling robots to directly map visual observations to control actions. Pretrained vision foundation models (VFMs) such as DINOv2 (Oquab et al., 2024), CLIP (Radford et al., 2021), and ViT (Dosovitskiy et al., 2020), provide transferable visual representations that support robotic perception and control with certain generalizability, improving the scalability of robotic systems (Huang et al., 2024; Wan et al., 2024).

However, executing even a single robotic task, and especially a diverse set of tasks, often requires multiple implicit visual competencies that a single VFM cannot fully capture. Directly integrating multiple VFMs for robot tasks increases computational and operational complexity. Previous works (Ranzinger et al., 2024; Shang et al., 2024; Chen et al., 2025) distill diverse foundation models into a unified representation, but three key challenges remain. First, heterogeneous VFM features are often misaligned, so a unified representation tends to dilute or discard model-specific capabilities. Second, the policy head must extract task-relevant information from the fused representation, which limits flexibility to leverage the most relevant VFMs across tasks and leads to suboptimal results. Third, existing distilled models typically require full retraining to incorporate robot-domain knowledge and it is hard to scale computation (down for simple tasks and up for complex tasks).

---

*Corresponding Author.

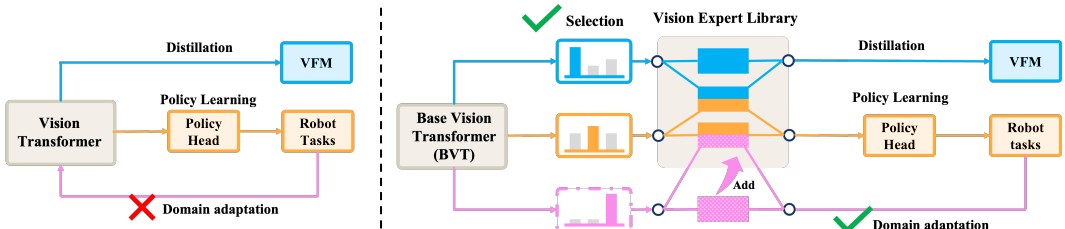

Figure 1: **A comparison between our VER and previous distillation framework.** Our method not only enhances knowledge distillation from vision foundation models (VFMs) into vision experts but also offers two key advantages over previous works (Ranzinger et al., 2024; Shang et al., 2024). First, VER trains a lightweight router that dynamically selects vision experts for downstream robot policies. Second, VER allows the integration of additional trainable experts, enabling the adaptation to robot-specific domain knowledge to further improve robotic performance.

To address these limitations, we propose **VER**, a **V**ision **E**xpert transformer for **R**obot learning via foundation distillation and dynamic routing. VER distills knowledge from multiple vision foundation models into a unified representation library and uses a dynamic routing mechanism to selectively activate the most relevant experts for robot policy learning. Specifically, VER introduces a Mixture-of-Experts (MoE)-based Vision Expert Library (VEL), replacing traditional static vision transformer backbones with a collection of specialized experts, each capturing distinct aspects of visual understanding. This design enables robots to selectively leverage the specialized experts best suited for task-aware policy learning.

Our method operates in three stages as shown in Fig. 1. First, during pretraining, we distill knowledge from multiple VFMs into a vision expert library using Teacher-Specific Routers with mutual-information regularization. This covers a broad spectrum of visual knowledge while maintaining efficiency via sparse expert activation. Second, in the robotic policy learning phase, we freeze all pretrained vision experts and fine-tune only a lightweight Robot Router that dynamically selects task-relevant experts, whose outputs are fed to a policy head to generate actions. To expand selection capacity across patches and layers, enhance exploration, and prevent premature convergence to suboptimal expert combinations, we employ *Patchwise Expert Routing* with *Curriculum Top-K Annealing*, leading to more robust policy learning. Third, we offer parameter-efficient fine-tuning strategies that scale expert utilization and facilitate the integration of robot-domain knowledge.

Across different types of policy heads, such as diffusion and flow matching policies (Chi et al., 2023; Zhang & Gienger, 2024), extensive experiments on diverse robotic benchmarks show that VER achieves state-of-the-art performance. With *Patchwise Expert Routing* and *Curriculum Top-K Annealing*, VER suppresses high-norm background outliers and reduces information in task-irrelevant patches while preserving details in task-critical regions, yielding more compact and discriminative visual features and robust policy learning.

## 2 RELATED WORKS

### 2.1 VISION FOUNDATION MODELS FOR REPRESENTATION

Vision Foundation Models (VFMs) have revolutionized computer vision through self-supervised and weakly-supervised learning on large-scale datasets (Radford et al., 2021; Caron et al., 2021; Oquab et al., 2024). Notable examples include CLIP (Radford et al., 2021) which pioneered image-text joint embeddings, DINOv2 (Oquab et al., 2024) which advanced self-supervised learning, and SAM (Kirillov et al., 2023) specialized for segmentation tasks.

Knowledge distillation has emerged as a powerful paradigm for transferring learned representations from large teacher models to more compact student architectures (Hinton et al., 2015; Romero et al., 2014). While traditional distillation approaches focus on compressing a single teacher into a smaller student (Hinton et al., 2015), recent advances have explored multi-teacher distillation (You et al., 2017; Shang et al., 2024) where complementary knowledge from multiple source models is combined. Theia (Shang et al., 2024) demonstrated that careful fusion of representations from diverse vision foundation models can achieve superior performance for downstream robotic tasks. However,

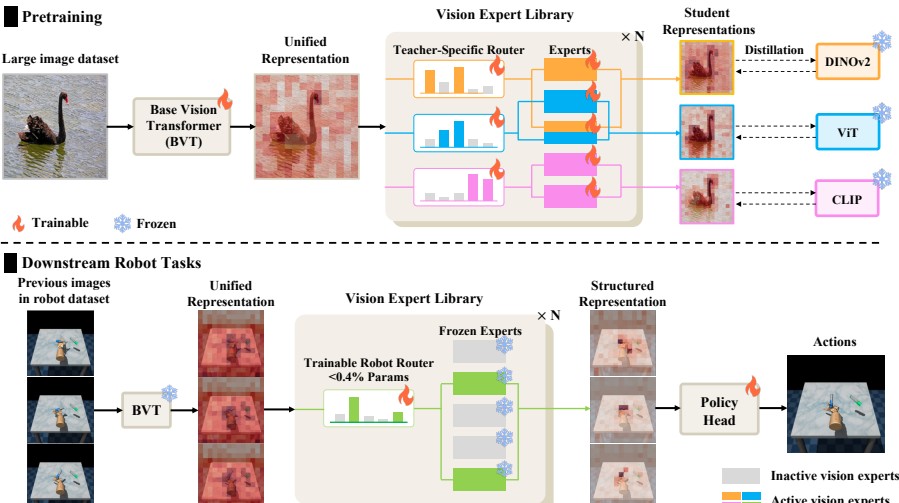

Figure 2: **Overall structure of VER.** VER comprises two key components: the Base Vision Transformer (BVT), which processes images into unified representations; the Vision Expert Library (VEL), which stores a diverse set of specialized vision experts and selectively utilizes the experts to mimic teacher vision foundation models and enhance performance in downstream robotic tasks. Our framework consists of two phases: (1) Pretraining, where we distill multiple foundation models (DINOv2 (Oquab et al., 2024), ViT (Caron et al., 2021), CLIP (Radford et al., 2021)) into VER; (2) Downstream Robotic Tasks, where we freeze the experts and train a lightweight Robot Router ($< 0.4\%$ parameters) that dynamically selects task-relevant visual features to guide the policy head in generating appropriate robotic actions. This two-stage approach enables efficient knowledge distillation from diverse vision foundation models and adaptive feature selection for robotic tasks.

these methods typically produce static representations with fixed weights, limiting their adaptability to specific downstream tasks.

Our work differs by distilling multiple VFMs into a specialized expert library rather than a single unified representation, preserving diverse knowledge from VFMs while enabling task-specific feature selection through learned routing mechanisms.

## 2.2 MIXTURE OF EXPERTS IN VISION AND POLICY

Mixture of Experts (MoE) architectures (Shazeer et al., 2017a; Fedus et al., 2022; Riquelme et al., 2021; Wang et al., 2024) have gained popularity for their ability to scale model capacity without proportional increases in computational costs by activating only a subset of expert networks for each input.

Router design represents a critical component in MoE systems, determining which experts process specific inputs. Top-$k$ routing (Shazeer et al., 2017a) selects the $k$ highest-scoring experts for each token, while Switch Transformers (Fedus et al., 2022) employ a simpler top-1 routing for efficiency. Recent work has explored learned routing mechanisms (Dai et al., 2022; Wang et al., 2024) that balance expert utilization while preserving specialization. Sparse MoE Router (Wang et al., 2024) introduced mutual information maximization between tasks and experts to encourage meaningful specialization while maintaining balanced utilization.

While MoE has been widely applied in language processing and general computer vision (Riquelme et al., 2021; Fedus et al., 2022), its application to robotic learning remains relatively unexplored. Our work bridges this gap by adapting MoE principles for vision-based robotic policy learning, introducing specialized routers that dynamically select visual representations most relevant to specific robotic tasks.

## 2.3 Visuomotor Robotic Policy Learning

Visuomotor robotic policy learning maps visual observations directly to robot actions (Levine et al., 2016; Brohan et al., 2023; Liang et al., 2024a; Mu et al., 2024; Chen et al., 2024), showing better generalization ability compared to state-based methods (Janner et al., 2022; Liang et al., 2023; Ajay et al.; Ni et al., 2023; Liang et al., 2024b). Recent approaches have leveraged pre-trained vision models to improve sample efficiency and generalization (Nair et al., 2023; Xiao et al., 2022; Radosavovic et al., 2023; Chen et al., 2025; Kim et al.; Black et al., 2024), but typically use fixed visual encoders that may not capture optimal representations for specific tasks. A persistent challenge is identifying which visual features are most relevant for different robotic tasks (Xiao et al., 2022; Luo et al., 2023). Current methods using attention mechanisms (Luo et al., 2023) or feature selection (Jiang et al., 2025) often lack the flexibility to incorporate diverse visual expertise from foundation models. Our work advances this field by introducing a dynamic visual representation selection mechanism specifically for robotic tasks. Unlike fixed visual feature approaches, ours enables selective leveraging of different representations from a diverse expert library based on task requirements, leading to more robust policies across varied robotic scenarios.

## 3 Method

### 3.1 Overview

In this section, we present our VER framework for visuomotor robot policy learning, as illustrated in Fig. 2. Our approach begins with visual perception, where input images are processed by a *Base Vision Transformer* (BVT) to extract foundational visual features, referred to as unified representations. These representations are then fed into a *Vision Expert Library* (VEL), a collection of specialized neural network experts designed to capture diverse aspects of visual understanding. A dynamic routing mechanism determines which experts should be activated based on the specific task: during pretraining, to mimic the teacher vision foundation models (VFMs); and for downstream robot tasks, to select experts that enhance performance. This mechanism enables selective attention to the most relevant visual features. Finally, the outputs of the selected experts are integrated to generate a more structured representation, which is either used to replicate the teacher VFMs or passed to a policy head that translates these representations into robot actions.

### 3.2 Model Architecture

As illustrated in Fig. 2, our approach consists of two main components: a *Base Vision Transformer* (BVT) that generates unified feature representations from input images, and a *Vision Expert Library* (VEL) comprising specialized experts that capture diverse visual representations from various vision foundation models.

We design VER based on a modified vision transformer (Dosovitskiy et al., 2020) architecture, where the FeedForward Network in the last $N$ transformer layers is replaced with Mixture of Experts (MoE) (Shazeer et al., 2017b) modules. The initial unaltered ViT layers are referred to as BVT, while the MoE-enhanced later layers constitute the VEL.

In the $n$-th MoE layer (Shazeer et al., 2017b) of VEL, where $n \in \{1, 2, ..., N\}$, we incorporate $L$ experts $\{\mathcal{E}_l^n\}_{l=1}^L$, each implemented as a multilayer perceptron (MLP). We then introduce Teacher-Specific (TS) Routers $\mathcal{R}_i^n$, where $i \in 1, 2, ..., I$ corresponds to each teacher vision foundation model. TS Router $\mathcal{R}_i^n$ takes in the input feature vector $x \in \mathbb{R}^{1 \times M}$ and learn a MLP to determine the score for each expert, as well as score noise. During inference, only the top-$K$ scoring expert networks are activated, while the remaining experts remain inactive, ensuring computational efficiency. The MoE output $y$ is computed as:

$$\begin{cases} y = \sum_{l=1}^L \mathcal{R}_i^n(x, l) \cdot \mathcal{E}_l^n(x), \\ \mathcal{R}_i^n(x, l) = m_l \cdot p_l, \quad p = \text{Softmax}(z), \quad z = s_1 + \epsilon, \\ [s_1; s_2] = \text{MLP}(x), \quad \epsilon \sim \mathcal{N}(0, \text{SoftPlus}(s_2)) \end{cases} \quad (1)$$

where $z$ is the noisy logit, $p_l$ is the routing probability for expert $l$, and $m_l \in \{0, 1\}$ is the Top-$K$ indicator (which equals 1 if $p_l$ is among the $K$ largest probabilities, and 0 otherwise). This sparse gating mechanism enables efficient computation while maintaining representation quality.

### 3.3 DISTILLATION TRAINING

Building on our network architecture, we now describe the distillation process that enables our model to acquire diverse visual representations from multiple VFMs, forming a comprehensive library of specialized vision experts for effective utilization in downstream robot tasks.

Given an input image $x$, the BVT $f(\cdot)$ first processes the image to produce a unified rich representation $z = f(x)$. Subsequently, the VEL $g(\cdot, \cdot)$ generates teacher-specific representations $y = g(z, i)$, where $i$ denotes the index of the specific teacher model being mimicked. Note that $g(\cdot, i)$ utilizes the corresponding TS Router to select the most appropriate experts for emulating the $i$-th VFM's representational characteristics.

Following (Shang et al., 2024), we formulate the distillation loss $\mathcal{L}_{distill}$ as a weighted combination of cosine and smooth L1 losses:

$$\mathcal{L}_{distill} = \sum_{i=1}^{I} \alpha_i [\beta \mathcal{L}_{cos}(h_i(g(f(x), i)), t_i(x)) \\ + (1 - \beta)\mathcal{L}_{sL1}(h_i(g(f(x), i)), t_i(x))], \tag{2}$$

where $h_i(\cdot)$ represents a projection head for the $i$-th teacher, $t_i(x)$ is the representation from the $i$-th teacher model, $\alpha_i = 1/I$ and $\beta = 0.9$.

To mitigate optimization conflicts and ensure balanced expert utilization when distilling multiple Vision Foundation Models (VFMs), we introduce a teacher-level mutual information loss (Wang et al., 2024). Unlike standard token-level load-balancing heuristics, this objective explicitly maximizes the mutual information between the categorical teacher variable $\mathcal{I}$ and the routed experts $\mathcal{E}^n$ across all MoE layers:

$$\mathcal{L}_{mi} = -\sum_{n=1}^{N} I(\mathcal{I}, \mathcal{E}^n) = -\sum_{n=1}^{N} \sum_{l=1}^{L} \sum_{i=1}^{I} p(\mathcal{I}_i, \mathcal{E}_l^n) \log \frac{p(\mathcal{I}_i, \mathcal{E}_l^n)}{p(\mathcal{I}_i)p(\mathcal{E}_l^n)}, \tag{3}$$

where we assume a uniform prior over the teachers, setting $p(\mathcal{I}_i) = \frac{1}{I}$.

To further analyze the underlying mechanism of this objective, we examine its entropy decomposition, $I(\mathcal{I}, \mathcal{E}^n) = H(\mathcal{E}^n) - H(\mathcal{E}^n \mid \mathcal{I})$, which intrinsically couples two critical regularization effects. First, maximizing the marginal entropy $H(\mathcal{E}^n)$ acts as a global load balancer; it encourages a uniform marginal expert-usage distribution, effectively preventing expert routing collapse from an information-theoretic perspective. Second, minimizing the conditional entropy $H(\mathcal{E}^n \mid \mathcal{I})$ enforces teacher-specific specialization by minimizing the uncertainty of expert assignment given a specific teacher $\mathcal{I}_i$. This promotes sparse, highly predictable routing trajectories, encouraging different VFMs to activate disjoint expert subsets. Consequently, it preserves the fine-grained visual semantics of individual teachers and minimizes gradient interference within the shared MoE pool. We defer further theoretical analysis and the formal definition of the routing probabilities to Appendix D.2.

The overall pre-training objective is thus formulated as:

$$\mathcal{L}_{pretrain} = \mathcal{L}_{distill} + \gamma \mathcal{L}_{mi}, \tag{4}$$

where $\gamma$ is a scaling hyperparameter empirically set to $0.0005$.

### 3.4 ROBOT POLICY TRAINING

After distilling diverse visual representations, we freeze a *Vision Expert Library* (VEL) together with a frozen base visual transformer (BVT) for downstream robot tasks. We introduce a lightweight robot router $\mathcal{R}_{robot}^n$ that selects task-relevant vision experts and feeds the resulting representations to a newly trained policy head to produce actions.

We consider two routing modes for robot tasks.

**Teacher Routing (TR)** Because pretrained vision foundation models perform strongly on robot tasks, one option is to choose which VFM to use by selecting among the Teacher-Specific (TS)

Table 1: **Per-task performance comparison (success rate in %) across various robotic benchmarks.** The same policy head as (Shang et al., 2024) is used for a fair comparison on vision encoder. The best result is in **bold** and the second best result is underlined. Our approach (VER) outperforms previous state-of-the-art methods across 11 diverse tasks from Franka Kitchen (Gupta et al., 2020), Meta-World (Yu et al., 2020), and Adroit (Rajeswaran et al., 2018) environments, achieving the highest average success rate (74.7%).

| Model | LightOn | DoorOpen | DoorSlide | KnobTurn | Microwave | BinPick | ButtonPress | DrawerOpen | Hammer | Pen | Relocate | Average |
|---|---|---|---|---|---|---|---|---|---|---|---|---|
| **VC-1** (Majumdar et al., 2023) | 1.6 | 0.2 | 14.4 | 1.2 | 1.8 | 66.7 | 56.0 | **100.0** | 93.3 | 68.0 | 24.0 | 42.6 |
| **MVP** (Xiao et al., 2022) | 13.6 | 5.3 | 17.8 | 1.8 | 4.0 | 73.3 | 82.7 | **100.0** | 97.3 | 77.7 | 26.7 | 48.7 |
| **R3M** (Nair et al., 2023) | **67.3** | 31.2 | 83.1 | 35.4 | 35.8 | 92.0 | 68.0 | **100.0** | 98.7 | 73.3 | 58.7 | 67.6 |
| **RADIO** (Ranzinger et al., 2024) | 35.2 | 19.7 | 69.2 | 24.4 | 25.3 | 82.7 | 80.0 | **100.0** | 100.0 | 66.7 | 45.3 | 61.3 |
| **VIP** (Ma et al.) | 61.3 | 25.2 | 83.0 | 44.6 | 31.3 | 70.7 | 76.0 | 98.7 | 96.0 | 73.3 | 29.3 | 62.8 |
| **Theia-B** (Shang et al., 2024) | 58.8 | 34.1 | 81.2 | 47.8 | 24.8 | 76.0 | 82.7 | **100.0** | 98.7 | 78.7 | 46.7 | 67.1 |
| **VER-B (Ours)** | 67.2 | **38.0** | **85.8** | **55.3** | **38.2** | **93.3** | **94.7** | **100.0** | 97.3 | **80.0** | **64.0** | **74.7** |

routers $\{\mathcal{R}_i^n\}_{i=1}^M$ learned during distillation. Specifically, for each image/frame $t$ and layer $n$, the robot router $\mathcal{R}_{\text{robot}}^n$ produces a categorical distribution $\boldsymbol{\pi}_{t,n} \in \Delta^{M-1}$ over $\{\mathcal{R}_i^n\}_{i=1}^M$. The selected TS router at layer $n$ is then used to select among the experts $\{\mathcal{E}_\ell^n\}_{\ell=1}^L$ in that layer. During training, we optimize the discrete teacher choice with the Gumbel–Softmax estimator:

$$\mathbf{z}_{t,n} = \text{softmax}\left(\frac{\log \boldsymbol{\pi}_{t,n} + \mathbf{g}}{\tau}\right), \qquad \mathbf{g} \sim \text{Gumbel}(0,1), \tag{5}$$

using a straight-through estimator, while at inference we take $\arg\max(\boldsymbol{\pi}_{t,n})$. We can share teacher logits across layers within a frame, i.e., $\boldsymbol{\pi}_{t,n} \equiv \boldsymbol{\pi}_t$, yielding a single teacher choice per frame; or allow per-layer logits $\boldsymbol{\pi}_{t,n}$ so that shallow and deep layers route to different teachers (e.g., DINOv2-like early, CLIP-like late) for finer control. We refer to the former as *Framewise Teacher Routing (FTR)* and the latter as *Layerwise Teacher Routing (LTR)*.

**Patchwise Expert Routing (PER)**  PER applies standard MoE routing *per patch token* (Eq. 1), offering maximal adaptivity to local content with minimal overhead ($< 0.4\%$ additional parameters). Because the router acts as a *planning selector* for task-relevant experts rather than enforcing balanced utilization, we omit the mutual-usage regularizer $\mathcal{L}_{\text{mutual}}$ (Eq. 3). However, naive end-to-end training without explicit regularization leads to premature convergence, which we formally identify as *early collapse*.

**Proposition 1** (Router Gradient in Top-K MoE). *Let $\mathcal{L}$ be the task loss. The gradient of the loss with respect to the router logit $z_l$ is given by:*

$$\frac{\partial \mathcal{L}}{\partial z_l} = p_l \left(m_l q_l - q\right) \tag{6}$$

*where $q_l := \left(\frac{\partial \mathcal{L}}{\partial y}\right)^\top \mathcal{E}_l$ is the expert-specific directional derivative and $q := \left(\frac{\partial \mathcal{L}}{\partial y}\right)^\top y$.*

As shown above, for any inactive expert ($m_l = 0$), the gradient simplifies to $\frac{\partial \mathcal{L}}{\partial z_l} = -p_l q$. This update is strictly independent of the expert's output $\mathcal{E}_l$ or potential contribution $q_l$. Because the shallow router adapts faster than the downstream policy network, initial random fluctuations permanently lock experts into an inactive state. This deprives them of informative gradient signals, causing the routing distribution to prematurely collapse into a suboptimal, seed-dependent configuration.

To mitigate this early collapse and encourage exploration, we introduce *Curriculum Top-K Annealing (CTA)*. We initialize training with all experts active ($K_0 = L$) and linearly anneal the number of active experts down to a target $K_{\min}$ over $S$ training steps:

$$K(s) = \max\left(K_{\min}, \left\lfloor L + 1 - (L + 1 - K_{\min})\frac{s}{S}\right\rfloor\right) \tag{7}$$

By applying CTA to PER's token-wise dispatch, we promote broad exploration in early stages and stable, sparse routing later, preserving inference efficiency at the target $K_{\min}$. Further analysis of router collapse and CTA is provided in Appendix B.

Table 2: **Per-task performance comparison (success rate in %) across different policy heads.** VER consistently outperforms Theia across ViLT (Kim et al., 2021), Flow-Matching (Zhang & Gienger, 2024) and Diffusion heads (Chi et al., 2023).

| Model | ViLT head | | Flow-Matching head | | | Diffusion head |
|---|---|---|---|---|---|---|
| | LIBERO | LIBERO-OOD | cross→bin | cube→cup | cylinder→plate | Real-world pour |
| Theia-T | 0.61 | 0.58 | 0.65 | 0.50 | 0.70 | 0.45 |
| VER-T | **0.70** | **0.71** | **0.95** | **0.75** | **0.85** | **0.90** |

Table 3: **Ablation of robot routing strategies.** Mean success rate $\pm$ standard deviation over 10 seeds. Best per task in **bold**; second best underlined. DINOv2, ViT, and CLIP denote the corresponding Teacher-Specific Routers. We can see different VFMs suit different tasks, and VER improves performance by dynamically routing to the appropriate experts distilled from these VFMs.

| Task | DINOv2 | ViT | CLIP | FTR | LTR | PER | PER+CTA |
|---|---|---|---|---|---|---|---|
| **pen** | 78.0±4.7 | 72.8±9.4 | 80.0±4.6 | **81.2±3.8** | 79.2±6.2 | 78.0±6.3 | 80.8±5.3 |
| **relocate** | 38.4±5.7 | 41.6±6.6 | 41.2±3.8 | 41.2±6.0 | 36.4±5.8 | 47.6±5.1 | **56.4±6.9** |

## 4 EXPERIMENTS

### 4.1 NETWORK STRUCTURE

To address the limited computational resources of robotic systems, we use DeiT-Tiny (Touvron et al., 2021) for VER-T, DeiT-Small for VER-S, and ViT-Base (Dosovitskiy et al., 2020) for VER-B. Distillation is performed on ImageNet-1K (Deng et al., 2009) from three foundation models—DINOv2 (Oquab et al., 2024), ViT (Caron et al., 2021), and CLIP (Radford et al., 2021). This configuration, aligned with Theia (Shang et al., 2024), controls for pretraining variations and enables a fair comparison. We use a total of $L = 6$ experts and activate $K = 2$ experts. To control complexity, VER replaces only the **last three layers** of a 12-layer transformer with the Vision Expert Library, yielding 9 standard transformer layers for the *Base Vision Transformer* and $N = 3$ MoE layers for the *Vision Expert Library*. Routing network is provided in Appendix C. Details of the pretraining procedure are provided in Appendix D.

### 4.2 PERFORMANCE ON ROBOT TASKS

With different policy heads such as ViLT (Liu et al., 2023; Kim et al., 2021), flow-matching and diffusion policy, we evaluate VER against pretrained vision encoders, including VC-1 (Majumdar et al., 2023), R3M (Nair et al., 2023), MVP (Xiao et al., 2022), RADIO (Ranzinger et al., 2024), VIP (Ma et al.), and Theia (Shang et al., 2024). Among these baselines, Theia is particularly strong as it distills multiple vision foundation models into a unified representation, while VIP leverages large-scale human video datasets to learn transferable features for robotic control.

We first follow Theia (Shang et al., 2024) and apply the same policy head across 11 diverse manipulation tasks spanning three benchmarks: 5 tasks from Franka Kitchen (Gupta et al., 2020) (LightOn, DoorOpen, DoorSlide, KnobTurn, Microwave), 4 from Meta-World (Yu et al., 2020) (Binpick, Buttonpress, DrawerOpen, Hammer), and 2 from Adroit (Rajeswaran et al., 2018) (Pen, Relocate). Second, we adopt the ViLT head (Liu et al., 2023; Kim et al., 2021) and evaluate VER on four LIBERO tasks (Liu et al., 2023), including LIBERO-OOD, where object colors are modified to test out-of-distribution generalization. Third, we apply a flow-matching head on three **Pick and Place** task in the Robomimic(Mandlekar et al., 2021). Finally, we use a diffusion policy head for the Pour task in real-world experiments.

As shown in Table 1, VER consistently outperforms prior approaches, achieving the highest average success rate of 74.7%. Furthermore, Table 2 shows that VER surpasses Theia across all policy heads both in simulation and real world experiments. Figure 5 shows the performance of our VER. Moveover, our VER also surpassed finetuned Vision-Language-Action model GR00T N1.5 NVIDIA et al. (2025) as shown in Table 10. More details and results can be found in Appendix E.

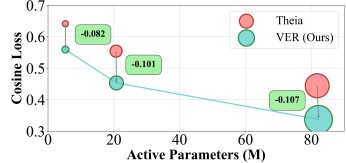 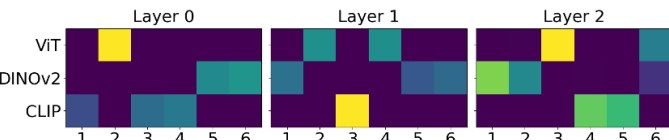

Figure 3: **Cosine loss for DI-NOv2 distillation.** Circle size indicates total parameters (TP).

Figure 4: **Expert utilization frequency across three MoE layers.** Heatmap shows how each teacher model activates experts (1–6) during distillation on ImageNet-1K.

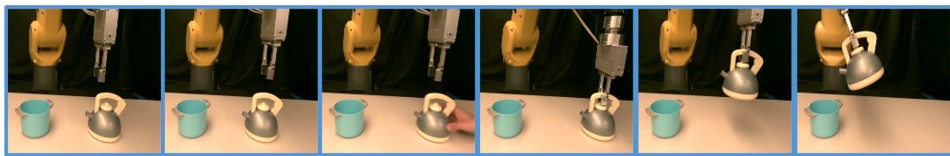

Figure 5: **Visualization of real world experiments.** We find with human interference (not in the training dataset), our VER can successfully complete the task.

### 4.3 DISTILLATION PERFORMANCE

Figure 3 shows that our framework effectively distills more knowledge from diverse foundation models; additional results are provided in Table 8. Figure 4 further illustrates expert utilization on ImageNet-1K (Deng et al., 2009). Instead of pre-assigning experts to teacher models, our Teacher-Specific Routers dynamically allocate them, with mutual information regularization encouraging diverse expert usage. We observe that ViT activates fewer experts, whereas DINOv2 and CLIP engage more, suggesting that ViT is easier to mimic while DINOv2 and CLIP present greater challenges. This trend is confirmed in Table 8, where the cosine loss after pretraining is significantly lower for ViT than for DINOv2 and CLIP. Overall, these findings demonstrate that our method outperforms fixed expert assignments by adaptively allocating more experts to stronger foundation models and fewer to weaker ones, thereby improving both utilization and distillation performance.

### 4.4 ABLATION ON ROUTER DYNAMICS IN ROBOT TASKS

In this subsection, we investigate the functional role of routers in robotic policy learning through three key experiments: (1) evaluating the impact of noisy gating on performance, (2) analyzing feature entropy evolution during training, and (3) examining the relationship between feature entropy and task performance. These experiments provide insights into how routers function as implicit planning modules for robotic tasks.

**Robot Routing Stratgies Performance.** We compare: (1) select one frozen Teacher-Specific (TS) Routers (DINOv2, ViT, or CLIP); (2) *Framewise Teacher Routing* (FTR) and *Layerwise Teacher Routing* (LTR); and (3) *Patchwise Expert Routing* (PER), with and without *Curriculum Top-K Annealing* (CTA). Results in Table 3 show that relying on a single VFM performs poorly across diverse tasks, whereas PER, when combined with CTA, adapts more effectively to local content across layers and achieves superior performance.

**Patch Feature Analysis.** To investigate the mechanism of CTA, we compute the norm of the last-layer patch features in VER, comparing models trained with and without CTA. As shown in Figure 6, CTA reduces high-norm outliers and concentrates attention on task-critical patches, whereas models without CTA exhibit large outliers in background regions. We further analyze patch features on 30% of the robot dataset by measuring entropy and mutual information before and after the *Vision Expert Library* (VEL). Patch features are first reduced to five dimensions via Principal Component Analysis (PCA), and then NPEET (Steeg, 2022) is used to estimate entropy and mutual information. As shown in Figure 7, PER+CTA filters out task-irrelevant background patches (lower mutual information before vs. after expert selection) while preserving task-relevant information (e.g., the target pen pose in pen-v0, which consistently appears in the left region of the image). Finally, Figure 8 compares feature norms from Theia, from VER before VEL, and from VER after VEL in the

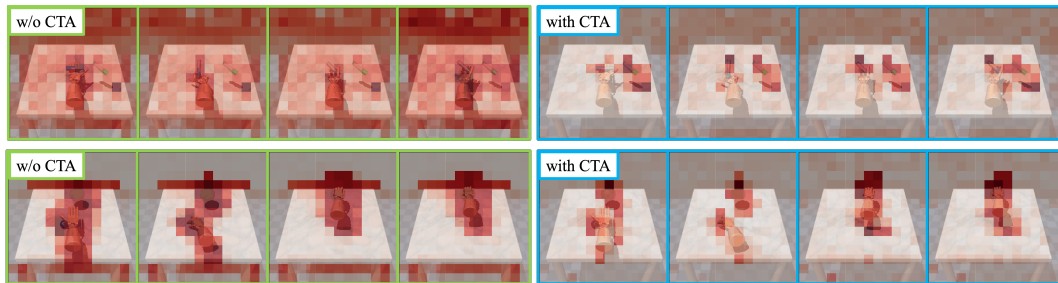

Figure 6: **Feature visualization of PER with and without CTA (seed = 0).** Row 1: *pen*; Row 2: *relocate*. Without CTA, the Robot Router attends broadly to the dexterous hand, objects, and task signals (e.g., target pen pose, target ball region). With CTA, the Robot Router suppresses task-irrelevant patches and concentrates on task-related, object-centric regions throughout execution.

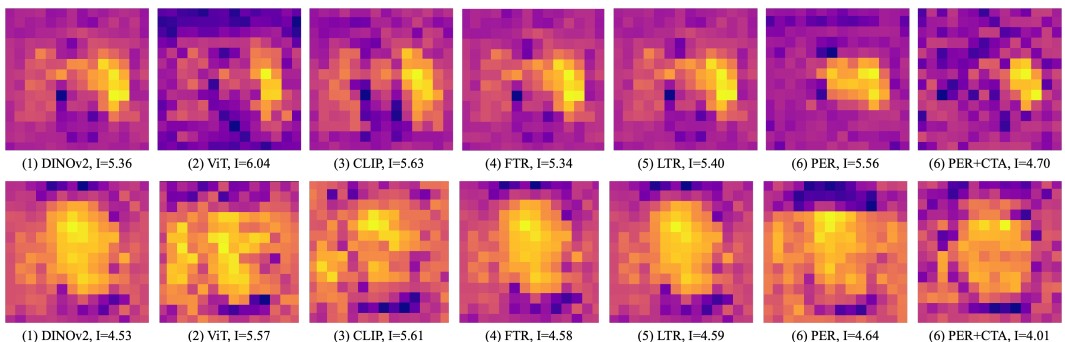

Figure 7: **Mutual information between patch features before and after the Vision Expert Library.** Row 1: *pen*; Row 2: *relocate*. PER+CTA suppresses information in background patches while preserving information in task-relevant regions, yielding a more compact visual representation (lower average per-patch mutual information). For example, in *pen*, the left-middle region containing the target pen pose exhibits higher mutual information.

**Pick and Place** task. This task consists of two stages: first pick up the cross, then place it into the bin. We supply patch features and robot proprioceptive state to the flow-matching policy head. We find that pretrained VFMs such as Theia, as well as VER before expert selection, broadly attend to all potentially important objects. After expert selection in VEL, however, the features focus exclusively on task-relevant objects (cross and bin) and suppress robot-related patches—consistent with our design choice to provide robot proprioceptive state directly to the policy, eliminating the need for robot-related information in patch features. More analysis can be found in Appendix E.3.

**Architectural Optimization and Depth.** To balance representational capacity with computational efficiency, we conduct a systematic ablation on expert granularity and architecture depth (See more in Appendix F). To ensure a rigorous comparison under a constant computational budget, we scale the MLP hidden dimension by $1/K$, maintaining a consistent active parameter count. As shown in Table 11, our analysis identifies the $(K, L) = (2, 6)$ configuration as an optimal trade-off, providing sufficient specialized knowledge without the routing overhead or optimization complexities associated with larger expert pools. Furthermore, while deeper MoE stacks yield lower pre-training distillation loss, they produce complex representations that are difficult for downstream policies to navigate, ultimately hindering task success (as shown in Table 12). Driven by these findings, VER employs a 12-layer transformer backbone where only the **last three layers** are replaced with the Vision Expert

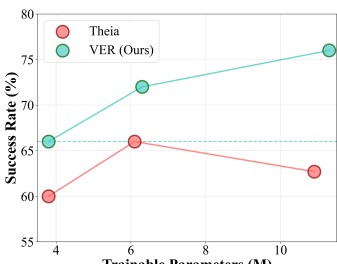

Figure 9: **Trainable parameters vs average success rates.** Performance is evaluated on *pen* and *relocate* tasks.

Figure 8: **Feature visualization compared with Theia (Shang et al., 2024) on** *place the cross into the bin*. Both Theia and our features *before* the Robot Router tend to attend broadly to other objects, the robot itself, and background regions, resulting in noisy feature norm. *After* routing, VER concentrates on task-relevant objects and suppresses robot-related and background patches.

Table 4: **Ablation study on** Top-K. More active experts lead to better performance.

| Model | TopK | AP(M) | Relocate | Pen | Avg |
|---|---|---|---|---|---|
| **Theia-Tiny** | - | 5.3 | 74.0 | 46.0 | 60.0 |
| **VER-Tiny** | 1 | 4.8 | 42.7 | 77.3 | 60.0 |
| | 2 | 5.2 | 52.0 | 80.0 | 66.0 |
| | 3 | 5.7 | 57.3 | 78.7 | **68.0** |

Table 5: **Ablation on the mixture of Distilled-Foundation-Model (DFM) and Train-from-Scratch (TFS) experts.**

| # DFM | # TFS | TopK | Relocate | Pen | Avg |
|---|---|---|---|---|---|
| 6 | 0 | 2 | 64.0 | 80.0 | 72.0 |
| 0 | 2 | 2 | 69.3 | 74.7 | 72.0 |
| 6 | 1 | 2 | **74.7** | **82.7** | **78.7** |

Library (VEL), establishing a configuration of $M = 9$ standard layers and $N = 3$ MoE layers that maximizes task performance.

**Inference Efficiency and Overhead.** Despite its enhanced capacity, VER maintains a minimal computational footprint. The inference time utilizing a diffusion policy in Table 2 on an RTX 4090 is 0.105s for both VER and Theia (Shang et al., 2024). While VER introduces lightweight routing components, it achieves substantially better downstream performance than Theia while utilizing fewer active and total parameters (see Table 8 and Figure 9). Ultimately, this demonstrates that VER delivers significant performance gains over existing baselines with comparable or even lower computational complexity.

**Scalability and Extensibility.** We examine the effect of the Top-K hyperparameter by finetuning only the lightweight Robot Router to adjust how many experts are selected per patch. As shown in Table 4, increasing the number of selected experts improves success rates but also increases computational cost. This demonstrates that VER enables a controllable trade-off between accuracy and efficiency without retraining the backbone or the experts. Beyond scalability, VER also offers extensibility by adaptive robot-domain knowledge integration. While distilled experts from pretrained VFMs encode strong general visual knowledge, they may miss information critical for specific downstream tasks. Our framework allows seamless integration of trainable experts tailored to such tasks. As shown in Table 5, adaptively combining Distilled-Foundation-Model (DFM) experts with Train-from-Scratch (TFS) experts achieves the best performance, highlighting the complementarity between generalist and task-specialized experts in enhancing overall task success.

## 5 CONCLUSION

In this paper, we introduce VER, a **V**ision **E**xpert transformer for **R**obot learning. Our approach distills knowledge from diverse vision foundation models into a vision expert library and employs a task-adaptive Robot Router to select task-relevant features for downstream control. To maximize selection capacity and prevent early collapse during router learning, we further propose *Patchwise Expert Routing* with *Curriculum Top-K Annealing*. Across multiple policy heads and a range of robotic benchmarks, VER achieves state-of-the-art performance. Patch-level analyses show that the Robot Router learns to selectively leverage pretrained experts, yielding increasingly structured representations that drive performance gains. In addition, VER is highly extensible: it seamlessly incorporates new robot-domain knowledge through expert addition, and scales the number of active experts to meet task complexity through lightweight router finetuning. These results highlight the value of expert-driven visual representation distillation and selection for robust, generalizable robot learning.

ETHICS STATEMENT

This research adheres to the ICLR 2026 Code of Ethics and upholds the principles of responsible research. Our experiments were conducted using publicly available datasets and self-collected data. No human subjects or vulnerable groups were involved, and no personally identifiable, sensitive, or harmful data were used in any part of this work. We have carefully considered the potential societal impacts of our methods, including risks of misuse or unintended consequences. We believe that our contributions primarily advance scientific understanding and do not pose foreseeable harm.

REPRODUCIBILITY STATEMENT

We follow the reproducibility guidelines outlined in the ICLR 2026 Author Instructions. To support reproducibility, we include detailed descriptions of dataset construction, model training, and evaluation in the main text and appendix. The main code and checkpoints are provided in the supplementary materials. Furthermore, we will release the complete source code, configuration files, and scripts on public platforms (e.g., GitHub and Hugging Face) to enable others to fully reproduce our results.

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

## A    LLM USAGE DISCLOSURE

We employed ChatGPT to assist with language refinement. All suggestions were reviewed and revised by the authors, who take full responsibility for the final manuscript.

## B    EARLY COLLAPSE OF ROUTER AND CTA

### B.1    ROUTER GRADIENT DYNAMICS IN TOP-K MOE

In this section, we analyze how the loss gradient with respect to the router logits $z_l$ is determined by the alignment between the MoE output $y$ and each **active** expert output $\mathcal{E}_l(x)$. We show that, for active experts ($m_l = 1$), the router updates compare the current mixture $y$ with each expert direction $\mathcal{E}_l(x)$ and increase the routing probability when moving towards that expert reduces the loss. In contrast, inactive experts ($m_l = 0$) receive only weak, indirect updates that do not depend on their outputs, so their re-entry into the Top-K set is not guaranteed, even if they could potentially improve the loss.

Recall Equation 1 here:

$$
\begin{cases}
y = \sum_{l=1}^{L} \mathcal{R}_i^n(x, l) \cdot \mathcal{E}_l^n(x), \\
\mathcal{R}_i^n(x, l) = \text{Top-K}(\text{Softmax}(s_1(x) + \epsilon), l), \\
[s_1(x); s_2(x)] = \text{MLP}_i^n(x), \quad \epsilon \sim \mathcal{N}(0, \text{SoftPlus}(s_2(x))).
\end{cases}
\tag{8}
$$

Let

$$
z_i^n(x, l) := s_1(x)_l + \epsilon_l,
$$

$$
p_i^n(x, l) := \text{Softmax}_l\big(z_i^n(x, \cdot)\big) = \frac{\exp(z_i^n(x, l))}{\sum_{j=1}^{L} \exp(z_i^n(x, j))},
\tag{9}
$$

Define the $\text{Top-K}(\text{Softmax}(s_1(x) + \epsilon), l))$ indicator $m_i^n(v, l)$ as

$$
m_i^n(x, l) = \begin{cases}
1, & \text{if } l \text{ is in Top-}K \text{ for token } x \text{ at layer } n, \\
0, & \text{otherwise.}
\end{cases}
\tag{10}
$$

Thus, the output $y$ is

$$
y = \sum_{l=1}^{L} m_i^n(x, l) p_i^n(x, l) \mathcal{E}_l^n(x)
\tag{11}
$$

For brevity, we temporarily suppress the indices $(n, i, x)$ and write $m_l, p_l, \mathcal{E}_l$. Denote $\mathcal{L}$ is the loss, then we compute the gradient with respect to the noisy logits $z_l, z_i^n(x, l)$ of expert $l$.

$$
\frac{\partial \mathcal{L}}{\partial z_l} = \left(\frac{\partial \mathcal{L}}{\partial y}\right)^{\top} \frac{\partial y}{\partial z_l}
$$

$$
= \left(\frac{\partial \mathcal{L}}{\partial y}\right)^{\top} \frac{\partial}{\partial z_l} \left(\sum_j m_j p_j \mathcal{E}_j\right)
\tag{12}
$$

$$
= \left(\frac{\partial \mathcal{L}}{\partial y}\right)^{\top} \sum_j \left(m_j \mathcal{E}_j \frac{\partial p_j}{\partial z_l}\right)
$$

$$
\begin{aligned}
\frac{\partial p_j}{\partial z_l} &= \frac{\partial}{\partial z_l} \left(\frac{e^{z_j}}{\sum_r e^{z_r}}\right) \\
&= \frac{e^{z_j} \delta_{jl} \sum_r e^{z_r} - e^{z_j} e^{z_l}}{\left(\sum_r e^{z_r}\right)^2} \\
&= \frac{e^{z_j}}{\sum_r e^{z_r}} \left(\delta_{jl} - \frac{e^{z_l}}{\sum_r e^{z_r}}\right) \\
&= p_j \big(\delta_{jl} - p_l\big).
\end{aligned}
\tag{13}
$$

Thus,

$$
\begin{aligned}
\frac{\partial \mathcal{L}}{\partial z_l} &= \left(\frac{\partial \mathcal{L}}{\partial y}\right)^\top \sum_j \left(m_j \mathcal{E}_j \frac{\partial p_j}{\partial z_l}\right) \\
&= \left(\frac{\partial \mathcal{L}}{\partial y}\right)^\top \sum_j \left(m_j \mathcal{E}_j p_j (\delta_{jl} - p_l)\right) \\
&= \left(\frac{\partial \mathcal{L}}{\partial y}\right)^\top \left(m_l \mathcal{E}_l p_l - p_l \sum_j m_j \mathcal{E}_j p_j\right) \\
&= p_l \left(\frac{\partial \mathcal{L}}{\partial y}\right)^\top (m_l \mathcal{E}_l - y) \\
&= p_l \left(m_l \left(\frac{\partial \mathcal{L}}{\partial y}\right)^\top \mathcal{E}_l - \left(\frac{\partial \mathcal{L}}{\partial y}\right)^\top y\right) \\
&= p_l (m_l q_l - q)
\end{aligned}
\tag{14}
$$

Where $q_l := \left(\frac{\partial \mathcal{L}}{\partial y}\right)^\top \mathcal{E}_l$, $q := \left(\frac{\partial \mathcal{L}}{\partial y}\right)^\top y$.

When $m_l = 1$, which is to say that expert $l$ is active, the gradient $\frac{\partial \mathcal{L}}{\partial z_l}$ effectively compares the performance between the current mixture output $y$ and the single expert output $\mathcal{E}_l(x)$, and determines the update direction for $z_l$. The change in $z_l$ will in turn shift the output $y$ through the routing probabilities. Let us assume that

$$
y' = y + \alpha\big(\mathcal{E}_l(x) - y\big),
$$

i.e., we slightly move the mixture output towards expert $l$ with a small step size $\alpha > 0$. Using a first-order Taylor expansion,

$$
\mathcal{L}(y') - \mathcal{L}(y) \approx \left(\frac{\partial \mathcal{L}}{\partial y}\right)^\top (y' - y) = \alpha \left(\frac{\partial \mathcal{L}}{\partial y}\right)^\top \big(\mathcal{E}_l(x) - y\big) = \alpha(q_l - q),
\tag{15}
$$

When $q_l < q$, moving $y$ towards $\mathcal{E}_l(x)$ reduces the loss, i.e., $\mathcal{L}(y') < \mathcal{L}(y)$ for small $\alpha$. In this case, the gradient $\frac{\partial \mathcal{L}}{\partial z_l}$ becomes negative, so $z_l$ increases under gradient descent. Since expert $l$ is active, a larger $z_l$ increases its routing probability and reinforces its selection. Thus, for active experts, the router automatically increases the weights of experts with smaller $q_l$ during training, leading to better expert assignments.

When $m_l = 0$, which is to say that expert $l$ is not active, there is no direct comparison between the current mixture output $y$ and the single expert output $\mathcal{E}_l(x)$ in the gradient signal: the update of $z_l$ does not involve $q_l$, and $\mathcal{E}_l(x)$ does not effectively participate in the gradient descent for this token. As a result, inactive experts receive only weak, indirect updates; consequently, their re-entry into the Top-K set is not guaranteed, even if they could potentially improve the loss.

## B.2  EARLY COLLAPSE OF ROUTER

In the previous section, we showed that only active experts ($m_l = 1$) receive expert-specific gradients based on the comparison between $y$ and $\mathcal{E}_l(x)$, whereas inactive experts ($m_l = 0$) receive only weak, indirect updates and their re-entry into the Top-K set is not guaranteed.

When training the robot router of VER for a robot task, a large, randomly initialized policy network $F_\theta$ is placed on top of $y$. Because the router is typically shallow, it can adapt much faster than this downstream network. In the early phase, random fluctuations of $q_l - q$ cause the router logits $z_l$ to concentrate on an essentially arbitrary subset of experts, meaning that expert selection is highly sensitive to random initialization and training seeds. Once a small set of experts is consistently selected as Top-K, the inactive experts ($m_l = 0$) receive only weak, indirect updates that do not depend on $\mathcal{E}_l(x)$, as discussed above. This *early router collapse* creates a mismatch in convergence rates: the router quickly commits to a suboptimal routing pattern determined by the random seed, while the large downstream network has not yet learned a meaningful representation. After this point, it is difficult for the router to revise its expert selection, because inactive experts no longer

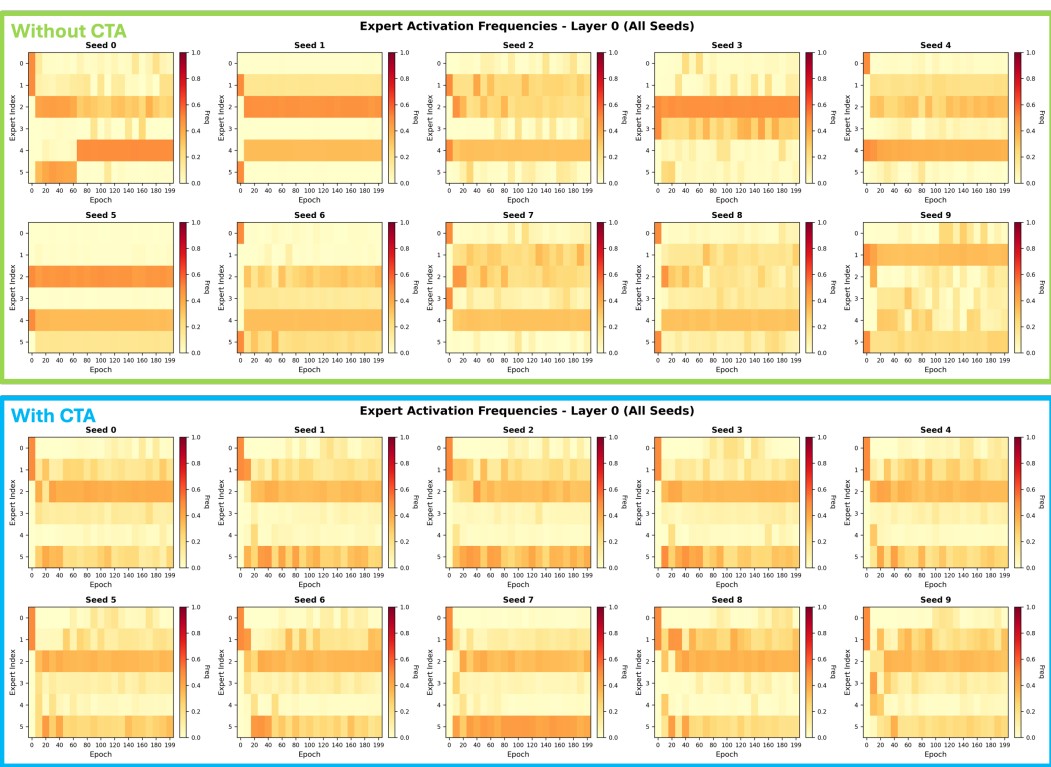

Figure 10: **Expert activation frequency during the training with and without CTA on *pen* (10 seeds).** Without CTA, expert activation frequencies converge prematurely in the early stage of training (for example, the activation frequencies of Seed 1 and Seed 5 do not change after 10 epochs) and depend strongly on the training random seed, leading to substantial variability across seeds. With CTA, the robot router explores more effectively and converges more consistently across different random seeds.

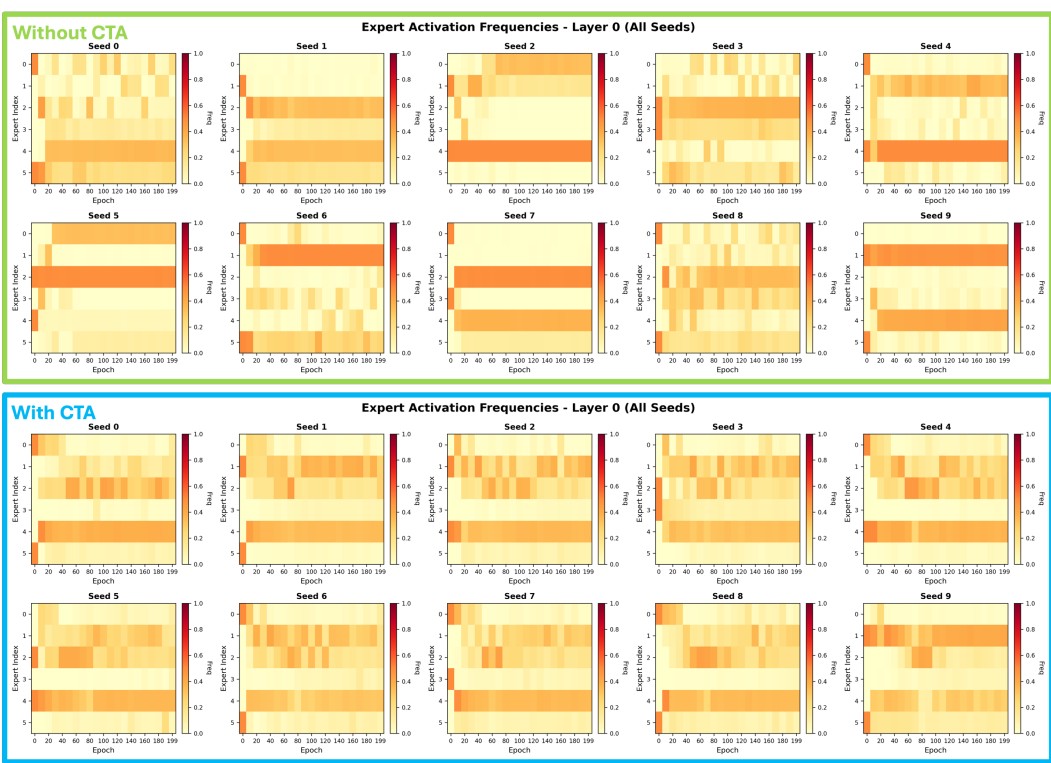

Figure 11: **Expert activation frequency during the training with and without CTA on** *relocate* **(10 seeds).** Without CTA, expert activation frequencies converge prematurely in the early stage of training (for example, the activation frequencies of Seed 5, 7 and 9 do not change after around 20 epochs) and depend strongly on the training random seed, leading to substantial variability across seeds. With CTA, the robot router explores more effectively and converges more consistently across different random seeds.

participate in gradient descent in a significant way, even if their fixed representations would later become beneficial as $F_\theta$ improves.

To address this issue, we propose *Curriculum Top-K Annealing (CTA)* in Equation 7, which initially activates all experts and then gradually decreases the number of active experts. This curriculum encourages exploration over expert combinations in the early stage and allows the policy network to converge, before enforcing a sparse Top-K routing pattern. As shown in Figure 10 and Figure 11, expert activation frequencies tend to converge early in training. With CTA, however, the expert activations exhibit a more consistent pattern across different random seeds. Figure 17 and Figure 18 present the final expert activation frequencies across all layers, further demonstrating that CTA leads to a more consistent final activation pattern across random seeds.

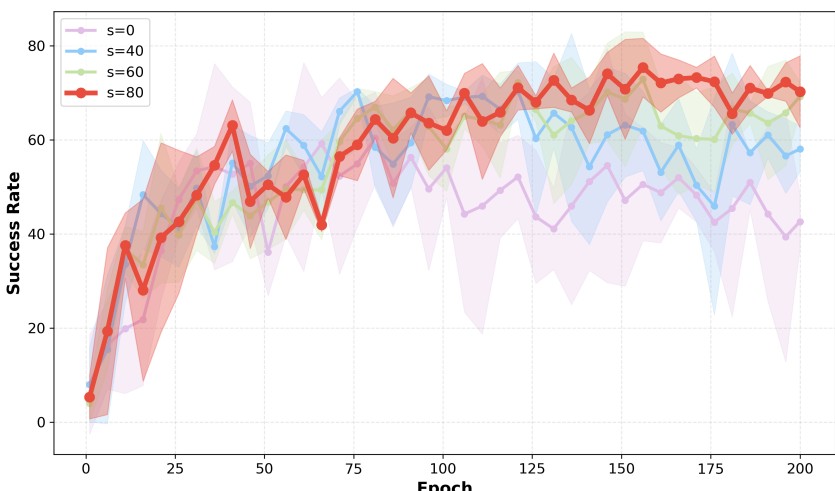

Figure 12: **Ablation study on $S$ in CTA on** *pen* **(3 seeds).** As $S$ increases, the success rate improves, and we also observe a reduction in the variance of the success rate, indicating more robust policy training.

### B.3 ABLATION STUDY ON $S$

In *Curriculum Top-K Annealing (CTA)* in Equation 7, there is a hyperparameter $S$ that controls the annealing schedule. A too small $S$ leads to insufficient exploration, whereas a too large $S$ results in slower convergence to the true $K_{\min}$ and thus higher computational cost. In practice, we choose $S$ based on the total number of training epochs. We evaluate performance on the pen-v0 and relocate-v0 tasks by training for 200 epochs with three random seeds and averaging the best success rate. Specifically, we set $S$ to 0, 40, 60 and 80 epochs, where $S = 0$ means that CTA is not applied. Table 6 shows that as $S$ increases, the success rate also increases. Moreover, Figure 12 shows that a higher value of $S$ reduces the variance of the success rate in the later stages of training, indicating more robust training with CTA.

Table 6: **Ablation study on $S$ in CTA on** *pen*.

| Task | S=0 | S=40 | S=60 | S=80 |
|------|-----|------|------|------|
| pen-v0 | 77.3±8.3 | 78.7±2.3 | **81.3±2.3** | **81.3±2.3** |

**Theorem 1** (Top-$K$ router gradients: active vs. inactive experts). *Consider a Top-$K$ MoE layer*

$$y = \sum_{j=1}^{L} m_j \, p_j \, \mathcal{E}_j(x), \qquad p_j = \mathrm{Softmax}_j(z), \qquad m_j \in \{0, 1\},$$

*and let $g := \frac{\partial \mathcal{L}}{\partial y}$. Define*

$$q_j := g^\top \mathcal{E}_j(x), \qquad q := g^\top y.$$

*Then for each expert logit $z_l$,*

$$\frac{\partial \mathcal{L}}{\partial z_l} = p_l \big( m_l q_l - q \big).$$

*In particular:*

- *If $m_l = 1$ (active), the router update depends on the expert-specific alignment $q_l$ and compares $\mathcal{E}_l(x)$ against the current mixture $y$; under gradient descent, $z_l$ increases iff $q_l < q$, i.e., iff infinitesimally moving $y$ toward $\mathcal{E}_l(x)$ decreases $\mathcal{L}$.*

- *If $m_l = 0$ (inactive), $\frac{\partial \mathcal{L}}{\partial z_l} = -p_l q$ is independent of $\mathcal{E}_l(x)$ (no expert-specific signal), so re-entry into Top-$K$ is not guaranteed even if $\mathcal{E}_l(x)$ could reduce the loss.*

**Theorem 2** (Early router collapse and curriculum Top-$K$ annealing (CTA)). *Let a shallow Top-$K$ router feed a much larger downstream network $F_\theta$ during early training. Because the router adapts faster, stochastic fluctuations in $(q_l - q)$ can rapidly concentrate probability mass onto an essentially arbitrary subset of experts, after which inactive experts receive no expert-specific gradients (Theorem 1). Consequently, the routing pattern becomes seed-sensitive and difficult to revise later, yielding* early router collapse. *A sufficient mitigation is* Curriculum Top-$K$ Annealing (CTA): *start training with all experts active ($K = L$) and gradually reduce $K$ to the target sparsity. This forces early exploration over expert combinations while $F_\theta$ learns meaningful features, and only later enforces sparse Top-$K$ routing, leading to more stable and consistent expert utilization across seeds.*

## C   ROUTING NETWORK

For the Teacher-Specific (TS) Router $\mathcal{R}_i^n$, we use a lightweight two-layer MLP (Linear $\rightarrow$ GELU $\rightarrow$ Linear) that produces per-patch logits over experts.

For the Robot Router $\mathcal{R}_{\text{robot}}^n$, we use different network structure for Teacher Routing and Patch Routing. For *Patchwise Expert Routing*, we adopt the same architecture as the TS Router. For *Framewise Teacher Routing* and *Layerwise Teacher Routing*, we first apply attention pooling over all patch features to obtain a summary token, followed by a three-layer MLP head (SiLU activations, dropout) that outputs logits over the Teacher-Specific Routers.

Table 7 shows the details of router network.

Table 7: **Network structure for routers.**

| Module | Input Granularity | Core Layers / Pooling | Output |
|---|---|---|---|
| TS Router | Single patch | Linear, GELU, Linear | Expert logits (per patch) |
| PER | Single patch | Linear, GELU, Linear | Expert logits (per patch) |
| FTR/LTR | All patches | **Attn** pooling; 3-layer MLP (SiLU, dropout) | Teacher logits (per patch/layer) |

## D   PRETRAIN EXPERIMENTS

### D.1   PRETRAINING DETAILS

We adopt DeiT-Tiny (Touvron et al., 2021) as the backbone for VER-T, DeiT-Small (Touvron et al., 2021) for VER-S, and ViT-Base (Dosovitskiy et al., 2020) for VER-B. We choose the last $N = 3$ layers as Vision Expert Library (VEL). The projection head $h_i(\cdot)$ consists of shallow CNNs,

following the same design of Theia (Shang et al., 2024). The training dataset is ImageNet-1K (Deng et al., 2009). We initialize the model weights using Theia and train VER on four A6000 GPUs for 50 epochs. The learning schedule consists of a linear warmup for the first 10% of training steps, followed by a constant learning rate of 0.002 for the next 40%, and then Cosine Annealing LR for the remaining steps.

## D.2  MUTUAL INFORMATION LOSS

In order to calculate $\mathcal{L}_{mi}$, we need to get $p(\mathcal{I}_i)$, $p(\mathcal{E}_l^n)$ and $p(\mathcal{I}_i, \mathcal{E}_l^n)$. Where $\mathcal{I}_i$ is the $i$th of teacher VFM, $\mathcal{E}_l^n$ is the $l$th expert at layer $n$. We assume each teacher model is equally important so $p(\mathcal{I}_i) = \frac{1}{I}$ where $I$ is the number of teacher models. For $p(\mathcal{I}_i, \mathcal{E}_l^n)$, we have

$$p(\mathcal{I}_i, \mathcal{E}_l^n) = p(\mathcal{I}_i)p(\mathcal{E}_l^n|\mathcal{I}_i) = \frac{1}{I}p(\mathcal{E}_l^n|\mathcal{I}_i)$$

$p(\mathcal{E}_l^n|\mathcal{I}_i)$ is the score of Teacher-Specific router $i$ for expert $l$ at layer $n$. Then $p(\mathcal{E}_l^n)$ can be calculated by $\sum_i p(\mathcal{I}_i, \mathcal{E}_l^n)$.

Intuitively, this mutual-information objective can be understood as follows. Maximizing $I(\mathcal{I}, \mathcal{E}^n)$ encourages the router to assign different experts to different VFMs while avoiding excessive overlap and promoting balanced expert utilization. Since

$$I(\mathcal{I}, \mathcal{E}^n) = I(\mathcal{E}^n, \mathcal{I}) = H(\mathcal{E}^n) - H(\mathcal{E}^n \mid \mathcal{I}), \tag{16}$$

maximizing $I(\mathcal{I}, \mathcal{E}^n)$ simultaneously increases $H(\mathcal{E}^n)$ and decreases $H(\mathcal{E}^n \mid \mathcal{I})$. A larger $H(\mathcal{E}^n)$ drives the marginal expert-usage distribution toward being approximately uniform, similar in spirit to traditional MoE token load-balancing losses. A smaller $H(\mathcal{E}^n \mid \mathcal{I})$ implies that, given a specific teacher $\mathcal{I}_i$ (and input $x$), the selected experts are more predictable and less uncertain. In practice, this encourages different teacher VFMs to rely on distinct subsets of experts, allowing them to preserve their fine-grained visual characteristics. Overall, the mutual-information loss both promotes task-agnostic load balancing across experts and preserves fine-grained vision features, thereby reducing conflicts when distilling multiple VFMs into a shared expert pool.

## D.3  PRETRAINING RESULTS

Table 8 demonstrates the distillation performance compared with Theia. We can see that our VER can achieve better distillation performance with similar active parameters. And VER-S with less active and total parameters achieve comparable distillation performance compared to Theia-B.

Table 8: **Distillation performance comparison between our VER and Theia across three foundation models (DINOv2, ViT, CLIP) using different loss metrics**. TP(M) denotes total parameters (Million), and AP(M) denotes active parameters (Million).

| Model | TP (M) | AP (M) | Cosine Loss | | | L1 Loss | | | MSE Loss | | |
|---|---|---|---|---|---|---|---|---|---|---|---|
| | | | DINOv2 | ViT | CLIP | DINOv2 | ViT | CLIP | DINOv2 | ViT | CLIP |
| **Theia-T** | 5.3 | 5.3 | 0.641 | 0.431 | 0.651 | 0.377 | 0.301 | 0.373 | 0.873 | 0.673 | 0.875 |
| **VER-T** (Ours) | 7.0 | 5.3 | 0.559 | 0.398 | 0.592 | 0.351 | 0.287 | 0.357 | 0.800 | 0.636 | 0.829 |
| **Theia-S** | 20.7 | 20.7 | 0.554 | 0.335 | 0.587 | 0.351 | 0.255 | 0.356 | 0.800 | 0.556 | 0.826 |
| **VER-S** (Ours) | 27.7 | 20.8 | 0.453 | 0.299 | 0.517 | 0.311 | 0.235 | 0.332 | 0.695 | 0.507 | 0.762 |
| **Theia-B** | 81.8 | 81.8 | 0.444 | 0.267 | 0.521 | 0.308 | 0.216 | 0.334 | 0.688 | 0.462 | 0.767 |
| **VER-B** (Ours) | 110.1 | 82.2 | 0.337 | 0.226 | 0.455 | 0.255 | 0.189 | 0.307 | 0.555 | 0.400 | 0.700 |

# E  ROBOT TASK EVALUATION

## E.1  BENCHMARKS & EVALUATION SETTINGS

**Franka Kitchen**  We mainly follow R3M (Nair et al., 2023) evaluation protocol. Specifically, we train the policy for 20,000 steps and evaluate success results every 1,000 steps throughout training. The final reported performance is based on the best average of three success rates observed during

evaluation. To ensure robustness, our results in each environment are averaged over different camera views (left and right) and different numbers of demonstrations (5, 10, and 25). We use the same policy network as Theia (Shang et al., 2024) for comparison. Specifically, we employ a three-layer MLP for CNN-based models using vector-based representations. For transformer-based models, we introduce a three-layer CNN before the MLP to process spatial inputs.

**Adroit & Meta-World**   We primarily follow the original evaluation setup of Cortex (Majumdar et al., 2023), with modifications to the training epochs for the Adroit environment. Since VER introduces additional training parameters for the router, which functions as a planning module requiring extended training, we increase the training epochs for the pen task to 200 and for the relocate task to 400, ensuring full performance convergence. As shown in Tab. 9, training for 100 epochs is insufficient for policy convergence. For VER-T, 200 epochs are enough for relocate. In this paper, our focus is primarily on the functionality of router, while optimizing its training efficiency is left for future work. We use the same policy network as in **Franka Kitchen**.

Table 9: **Performance vs. epoch.** We report average / highest success rates.

| Epoch | relocate | pen |
|---|---|---|
| 100 | 48.0/52.0 | 78.7/80.0 |
| 200 | 50.7/52.0 | 80.0/84.0 |
| 300 | 56.0/60.0 | –/– |
| 400 | 64.0/68.0 | –/– |

**LIBERO**   We select first four tasks from LIBERO_OBJECT (Liu et al., 2023) and train a ViLT (Kim et al., 2021) policy for 30 epochs, following the LIBERO evaluation protocol. In addition, we change colors for all the colors to evaluate performance in an out-of-distribution setting.

**Pick and Place**   We set up the *Pick and Place* task in robomimic (Mandlekar et al., 2021). The object is one of {cross, cube, cylinder}, and the container is one of {bin, cup, plate}. Objects are randomly positioned and oriented on the left side of the desk; containers are randomly positioned and oriented on the right side. We use a SpaceMouse to teleoperate the robotic arm in robomimic, collecting 50 human demonstrations, and then use MimicGen (Mandlekar et al., 2023) to generate 450 additional demonstrations, yielding 500 demonstrations per task. We train the flow-matching policy for 16,000 steps and evaluate over 40 trials. The flow-matching policy network is U-Net–based.

**Real-World Experiment**   We conduct real-robot experiments on a FANUC LR Mate 200iD/7L robotic arm equipped with an SMC gripper. The task is to pick up a teapot and pour into a cooking pot. Both the teapot and the cooking pot are randomly positioned and oriented. We collect 20 demonstrations, train a diffusion policy for 120,000 steps, and evaluate over 20 trials. The diffusion policy network is U-Net–based.

**Comparison with Vision-Language-Action Models**   We compare our VER model, utilizing a flow-matching policy head, against two primary baselines: (1) a Theia-based model equipped with a flow-matching policy, and (2) the fine-tuned vision-language-action (VLA) model, GR00T N1.5 (NVIDIA et al., 2025).

For each task, the models are trained using 500 expert demonstrations. Following the officially recommended fine-tuning protocol for the GR00T architecture, we fine-tune GR00T N1.5 for 20k steps with a batch size of 16. As shown in Table 10, our method consistently outperforms the baselines across all manipulation scenarios.

Table 10: **Performance comparison between VER and baselines.**

| Model | cross→bin | cube→cup | cylinder→plate |
|---|---|---|---|
| Theia | 0.65 | 0.50 | 0.70 |
| GR00T N1.5 (NVIDIA et al., 2025) | 0.75 | 0.73 | 0.70 |
| **VER (Ours)** | **0.95** | **0.75** | **0.85** |

Pick cube and place into cup

Pick cylinder and place into plate

Pick cross and place into bin

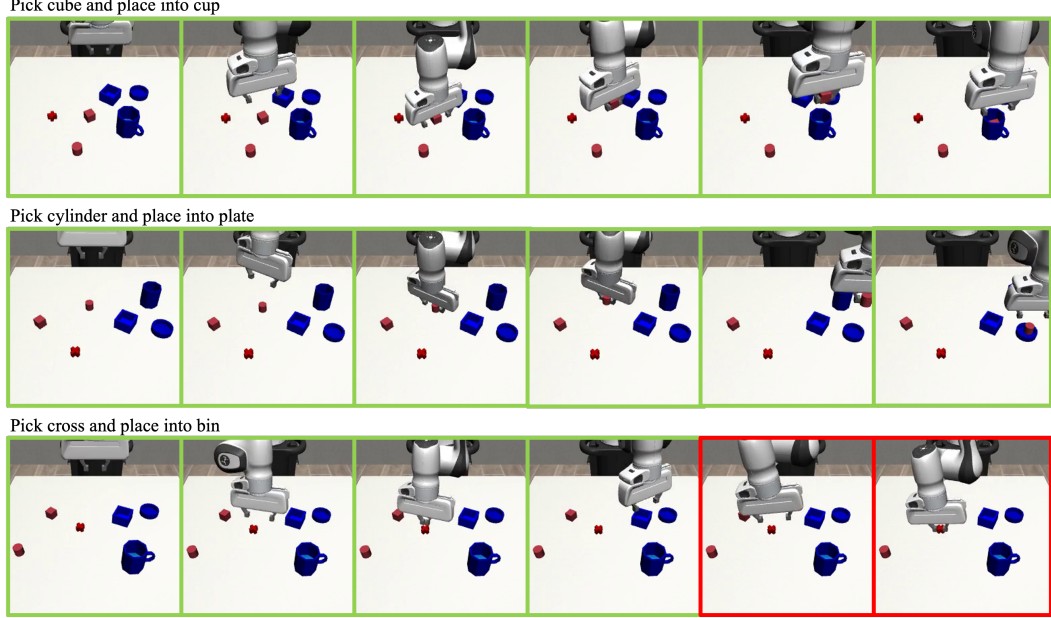

Figure 13: **Performance on *Pick and Place*..** We find that when the first attempt fails, the VER-equipped policy can retry and complete the task, as shown in the images with red boundaries.

### E.2 EXPERIMENT VISUALIZATIONS

Figure 13 shows the performance of VER in *Pick and Place* tasks. We find that when the first attempt fails, the VER-equipped policy can retry and complete the task, as shown in the images with red boundaries.

### E.3 PATCH FEATURE ANALYSIS

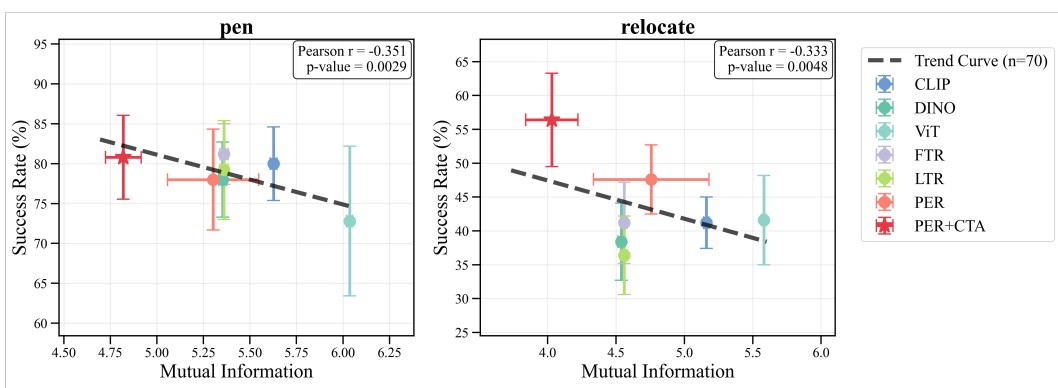

Figure 14: **Mutual information vs. success rate.** For each method (CLIP, DINOv2, ViT, FTR, LTR, PER, PER+CTA), we report the mean ± s.d. over 10 random seeds. The dashed line is an ordinary least-squares fit on 70 points (7 methods × 10 seeds) summarizing the overall trend.

Figure 14 shows that PER+CTA occupies the region with the lowest mutual information and the highest success rate. We fit a linear model to all 70 training results (7 methods × 10 random seeds) and observe a negative association: lower mutual information correlates with higher success. For the effect of CTA, we find that adding CTA markedly reduces both mutual information and variance across seeds, leading to more stable and robust training. Additional per seed visualizations with and without CTA (Figure 15) further support this observation: without CTA, the mutual information distribution varies substantially across different random seeds; with CTA, the Robot Router always

concentrates on the task relevant region (the left middle area with high mutual information corresponding to the target pen pose) and suppresses background regions with low mutual information.

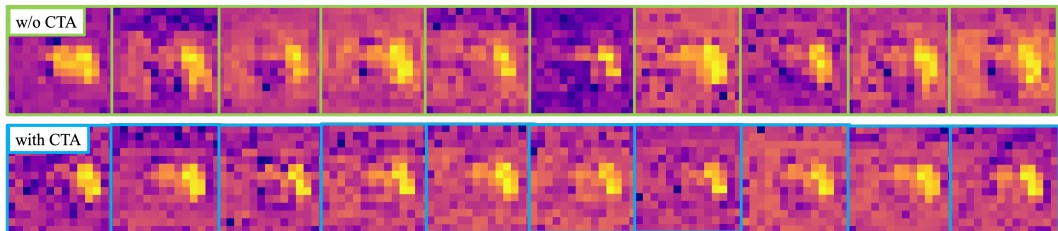

Figure 15: **Mutual information between patch features before and after the Vision Expert Library on *pen*.** We plot the image across 10 random seeds for training.

Figure 16 shows that without CTA, the Robot Router produces noisy patch embeddings with extremely large feature norms in background regions. Although it can sometimes attend to the correct regions and ignore task-irrelevant patches, its behavior is highly sensitive to the training random seed. This indicates that training a lightweight router is prone to early collapse and limited exploration. In contrast, with CTA, the Robot Router consistently focuses on task-relevant patches in a more robust manner. To further analyze this phenomenon, we plot the expert utilization frequency over the entire robot dataset for 10 random seeds in Figures 17 and 18. With CTA, the utilization frequencies are noticeably more consistent across seeds, which indicates that CTA helps avoid early collapse and insufficient exploration, thereby leading to more robust Robot Router learning.

# F    ABLATION ANALYSIS: MoE CONFIGURATION AND ARCHITECTURE DEPTH

We systematically evaluate the architectural design of our framework by analyzing the trade-offs between expert granularity, total model depth, and inference efficiency. To ensure a rigorous comparison under a constant computational budget, we scale the MLP hidden dimension by $1/K$ for all configurations, thereby maintaining a consistent number of active parameters per forward pass.

## F.1    EXPERT CONFIGURATION: ANALYSIS OF $K$ AND $L$

We first investigate the impact of the number of active experts ($K$) and total expert pool size ($L$). Models are pre-trained on a 25% subset of ImageNet-1K and evaluated on the pen-v0 task.

As summarized in Table 11, performance scales significantly from $(K, L) = (1, 3)$ to $(2, 6)$, followed by a marginal degradation at $(4, 12)$. This suggests that $(2, 6)$ represents a "sweet spot" in representational capacity; while increasing $L$ enhances the model's specialized knowledge, excessive experts may introduce routing instabilities and optimization complexities that hinder downstream policy performance.

Furthermore, we profile the inference latency of the Base Vision Transformer (BVT) against our MoE variants. We decouple the latency contributions of the router and the expert layers. As shown in Table 11, latency scales positively with $K$ and $L$, primarily driven by the increased overhead in the routing mechanism and expert computation. The $(2, 6)$ configuration is selected as our primary backbone, as it maximizes success rate with only a marginal increase in total latency.

Table 11: **MoE Configuration Analysis.** Success rates (%) are averaged over five random seeds; latency (ms) is measured with $batch\_size = 1$ on a VER-Tiny hardware platform.

| $K$ | $L$ | Success Rate (%) | Total Latency | Router | Expert |
|---|---|---|---|---|---|
| 1 | 3 | $44.0 \pm 11.9$ | **1.4491** | **0.5620** | **0.5464** |
| 2 | 4 | $48.8 \pm 5.3$ | 1.4873 | 0.5526 | 0.6021 |
| 2 | 6 | $\mathbf{69.6 \pm 4.8}$ | 1.6227 | 0.5855 | 0.6939 |
| 4 | 12 | $66.4 \pm 4.1$ | 1.9383 | 0.5847 | 1.0089 |

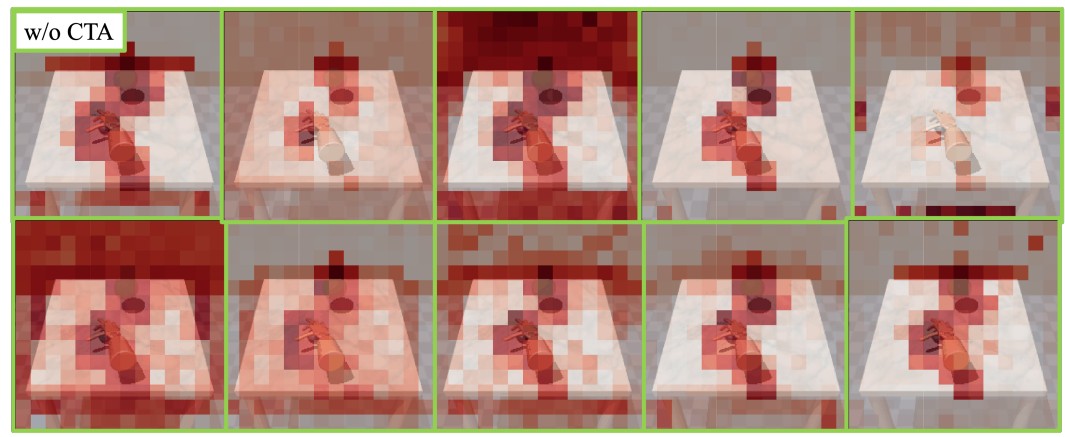

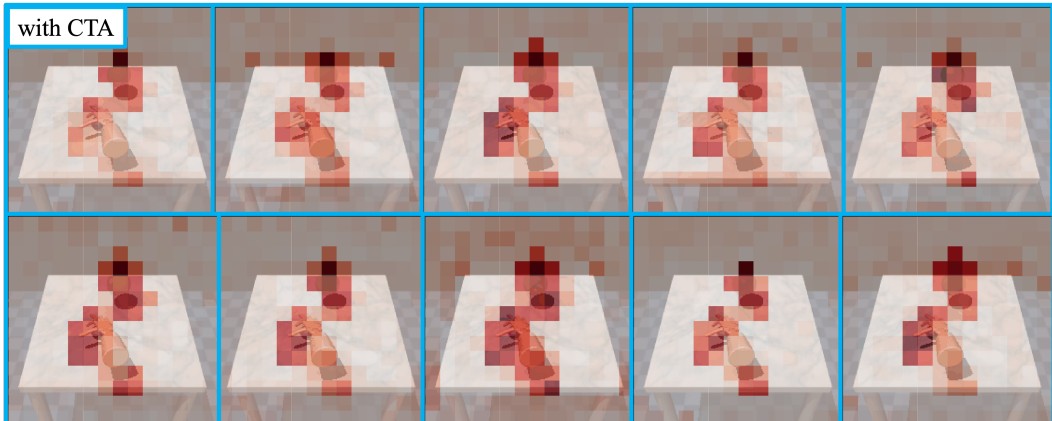

Figure 16: **Feature visualization of PER with and without CTA across 10 seeds.** Without CTA, the Robot Router attends broadly to the background and generates extreme feature-norm outliers, and its behavior is strongly influenced by the training random seed. With CTA, the Robot Router robustly suppresses task-irrelevant patches and concentrates on task-related regions across all the seeds.

## F.2 ARCHITECTURE DEPTH: BALANCING BVT AND VEL LAYERS

VER contains $M$ standard transformer layers (BVT) and $N$ sparse MoE layers (VEL). We study on $(M, N)$ to identify the optimal balance between general feature extraction and expert-driven specialization.

Results in Table 12 indicate that increasing the BVT depth from $M = 7$ to $M = 9$ improves both the distillation cosine loss and the final success rate. However, increasing the VEL depth to $N = 5$ presents a notable divergence: while it achieves the lowest distillation loss, the downstream success rate drops significantly to $48.8\%$. This suggests that while deeper MoE stacks are superior at mimicking pre-training targets, they may produce high-dimensional representations that are more difficult for the downstream policy to navigate. Consequently, we fix $N = 3$ to maintain a robust bridge between pre-training quality and task execution.

Table 12: **Ablation Study on Architecture Depth.** Comparison of standard layers ($M$) versus MoE layers ($N$).

| $M$ | $N$ | **Distill Cosine Loss** | **Success Rate (%)** |
|---|---|---|---|
| 7 | 3 | 0.561 | $50.4 \pm 12.5$ |
| 9 | 3 | 0.551 | $\mathbf{69.6 \pm 4.8}$ |
| 9 | 5 | **0.546** | $48.8 \pm 12.2$ |

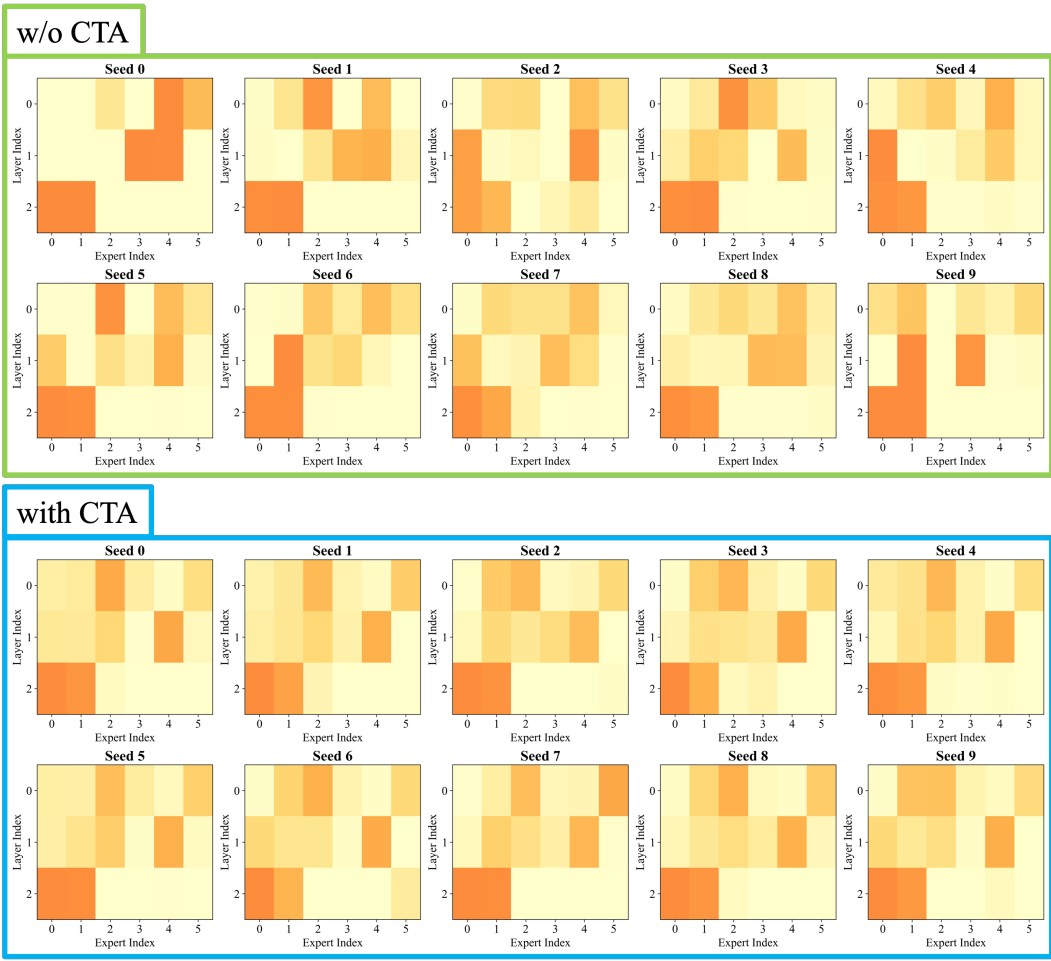

Figure 17: **Expert utilization frequency with and without CTA on *pen* (10 seeds).** Without CTA, expert utilization frequency varies substantially across random seeds. With CTA, it is more consistent across seeds, indicating improved training robustness and a more stable Robot Router.

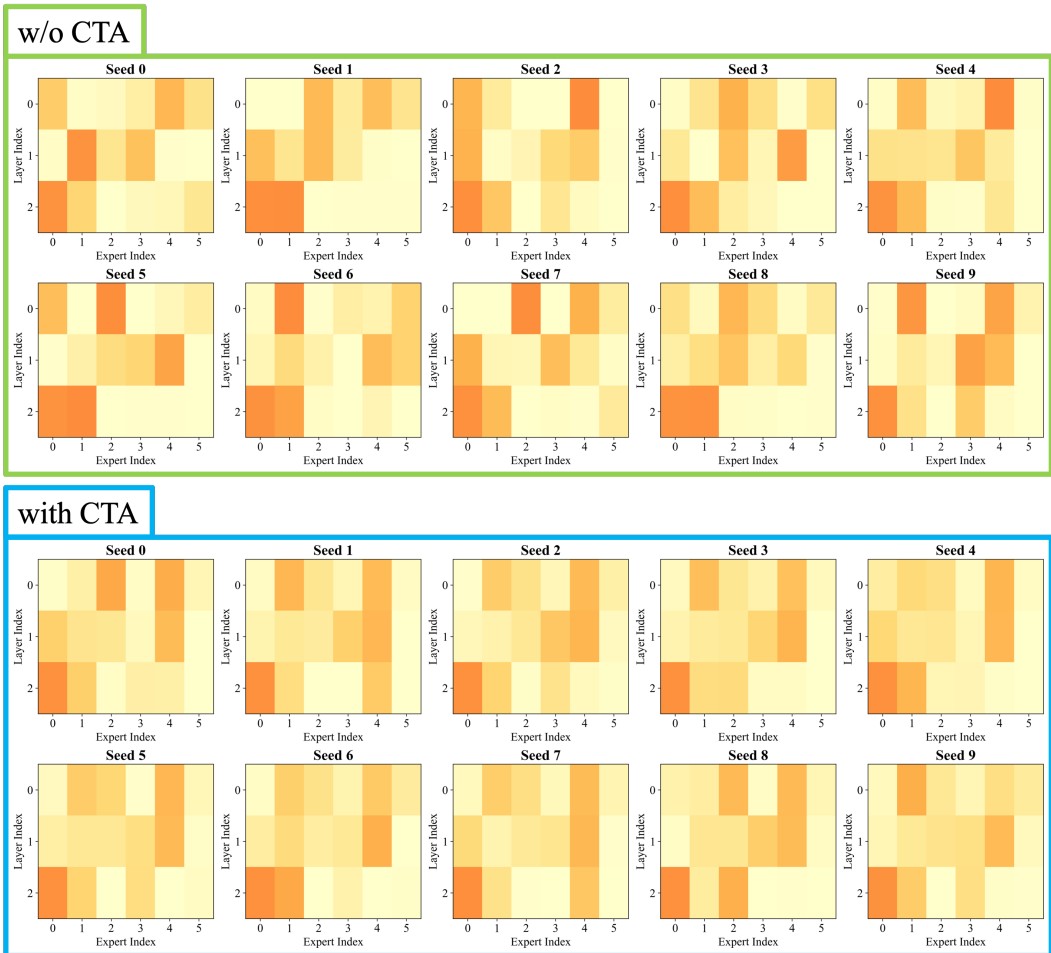

Figure 18: **Expert utilization frequency with and without CTA on *relocate* (10 seeds).** Without CTA, expert utilization frequency varies substantially across random seeds. With CTA, it is more consistent across seeds, indicating improved training robustness and a more stable Robot Router.

