# OpenReview forum: "VER: Vision Expert Transformer for Robot Learning via Foundation Distillation and Dynamic Routing"
_ICLR.cc/2026/Conference — ICLR 2026 Poster_

### Official Review · Reviewer_b4k1 · 2025-10-27

**Soundness:** 2
**Presentation:** 3
**Contribution:** 1
**Rating:** 2
**Confidence:** 4

**Summary:**

The paper introduces VER (Vision Expert Transformer for Robot Learning), an approach that consolidates Mixture-of-Experts (MoE) for downstream control tasks. The authors introduced a two stage approach, a routing network to select task relevant experts and a curriculum top-k annealing to improve the dynamic expert selection. The paper reports state-of-the-art performance on multiple robotics benchmarks.

**Strengths:**

- The paper demonstrates strong empirical results across a large and diverse set of benchmarks
- VER is an important practical contribution to real-world robot deployment and scalability

**Weaknesses:**

- The proposed mechanism is an important contribution to robotic learning, nevertheless the papers lacks theoretical justification and several design choices seem heuristic engineered. Further principled analysis and justification are needed.

- The ablation study is insufficient to justify several hyper-parameter choices and to justify the complexity of the proposed technique. My major concern here is it fails to isolate the performance gains attributed to each novel mechanism versus the underlying MoE architecture or increased model capacity.

**Questions:**

- What are the theoretical guarantees or formal analyses regarding how the annealing schedule in CTA impacts expert load balancing or the stability/convergence of the MoE router?

- Beyond empirical performance, what is the formal algorithmic advantage of PER over a standard token-wise or layer-wise routing scheme in the context of dense visual representations?

- The distillation loss uses mutual-information regularization. Please elaborate on the theoretical motivation for this specific regularization in this context.

---

> ### Author Response · Authors · 2025-11-24
>
> Thank you for highlighting the strong empirical performance and the practical significance of our work. We sincerely appreciate your detailed and constructive feedback. Below, we provide our responses to your comments.
>
> **Weakness:** The proposed mechanism is an important contribution to robotic learning, nevertheless the papers lacks theoretical justification and several design choices seem heuristic engineered. Further principled analysis and justification are needed.
>
> **Answer:** Thank you for recognizing the importance of our work for robot learning. In the revised version, we have substantially strengthened the theoretical analysis to better justify our design choices and clarify the underlying mechanisms. Specifically, we provide three complementary analyses:
>
> 1. **Theoretical analysis of router training and the motivation for CTA.**
>    We add a **formal analysis of the router gradients in Appendix B**. For active experts, the router gradient induces an informative reweighting of expert outputs: if the output of a selected expert leads to a lower loss, its routing score will be increased after the gradient update. In contrast, the router gradient for *inactive* experts does not contain information about those experts’ contributions, so the update direction for their routing scores is largely unconstrained and not guaranteed to be meaningful. This becomes particularly problematic when training a robot policy with a large, randomly initialized policy network, as it can lead to unstable routing behavior and collapse. Based on this analysis, we identify the core issue as arising from the treatment of inactive experts and therefore propose CTA to explicitly address this problem. We also report the expert activation frequency over the course of training, which empirically validates both the router-collapse issue and the effectiveness of CTA in mitigating it. Additional details are provided in Appendix B.
>
> 2. **Algorithmic advantage of Patch-wise Expert Routing (PER) over Teacher Routing (TR).**
>    We also analyze the advantage of PER from an algorithmic perspective. PER operates at the patch level and can explore a much larger combinatorial space of expert assignments than Teacher Routing. Consequently, it has the potential to find more optimal expert combinations. In contrast, TR searches only over a restricted subset of PER’s search space (e.g., by selecting among teacher-specific routers), which is theoretically more likely to yield suboptimal configurations. This comparison clarifies that PER is not a heuristic variant of TR, but a strictly more expressive routing scheme.
>
> 3. **Principled motivation for the mutual information loss.**
>    We further elaborate on the motivation for using a mutual-information-based loss instead of a standard token load-balancing loss. Our goal is twofold: (i) to preserve fine-grained visual features from the VFMs and (ii) to ensure that expert selection reflects task semantics. When distilling from multiple VFMs, their embedding spaces can differ significantly, which may lead to gradient conflicts if we naively combine their supervision. For better explanation, we analyze what is optimized when maximizing the mutual information between the expert selection and the teacher/task signal. Let $T$ denote the teacher (or task semantics) and $E$ the expert index. Maximizing  $I(T; E) = H(T) - H(T \mid E)$ encourages the teacher signal $T$ to be predictable given the expert choice $E$. Intuitively, each expert is then used by as few teachers as necessary, promoting specialization with respect to particular VFMs and reducing gradient conflict. As a result, experts become more semantically specialized, and the routing decisions are better aligned with the structure of the underlying vision features.
>
> We hope this response clarifies that our design choices are not ad hoc heuristics, but are grounded in theoretical considerations and in the concrete optimization issues that arise in robot learning.

---

> > ### Author Response · Authors · 2025-11-24
> >
> > **Weakness:** The ablation study is insufficient to justify several hyperparameter choices and the overall complexity of the proposed technique. My major concern is that it fails to isolate the performance gains attributable to each novel mechanism, as opposed to those arising from the underlying MoE architecture or increased model capacity.
> >
> > **Answer:** Thank you for this insightful comment. In the revised version, we have added more extensive ablation studies to better justify our hyperparameter choices and to disentangle the contributions of each component of our method.
> >
> > First, we include an additional ablation on the hyperparameter $S$ in CTA (see Appendix B.3), which clarifies its influence on performance.
> >
> > Second, we explicitly isolate the performance gains due to the MoE architecture and the proposed routing mechanisms. In particular, we compare our VER model with Theia, since Theia is also distilled from the same training dataset and teacher VFMs. The main architectural difference between Theia and VER is that, in VER, we replace the last three layers with an MoE structure and train an additional robot router for the downstream robot tasks. As shown in the following table, introducing the MoE structure, which better distills fine-grained VFM features (lower distillation loss; see Table 8), leads to consistent performance improvements over Theia. Moreover, adding CTA further improves performance.
> >
> > **Table:** **Ablation study on each component of VER.**
> >
> > |                     | **Theia**         | **VER**              | **VER+CTA**             |
> > |---------------------|-------------------|----------------------|-------------------------|
> > | **SR on pen**       | $74.0 \pm 2.8$    | $78.0 \pm 6.3$       | **$80.8 \pm 5.3$**      |
> > | **SR on relocate**  | $46.0 \pm 2.8$    | $47.6 \pm 5.1$ | **$56.4 \pm 6.9$**      |
> >
> > Figure 9 also shows that VER-Tiny outperforms Theia-Tiny and achieves performance comparable to Theia-Small and Theia-Base. This suggests that, while the increased capacity from the MoE structure is beneficial, the robot routing mechanism itself plays a crucial role in improving performance on robot tasks.
> >
> > Finally, to address concerns about complexity, we report the number of trainable parameters involved in robot policy learning (policy network + robot router). As shown below, the robot router constitutes only a small fraction of the total trainable parameters, indicating that the additional complexity introduced by our method is modest:
> >
> > **Table:** **Trainable parameters for policy learning with VER-T, VER-S, and VER-B.**
> >
> > |                                      | **VER-T** | **VER-S** | **VER-B** |
> > |--------------------------------------|-----------|-----------|-----------|
> > | **Total trainable parameters (M)**   | $3.81$    | $6.25$    | $11.31$   |
> > | **Robot router parameters (M)**      | $0.03$    | $0.11$    | $0.45$    |
> >
> > We hope these new ablations and parameter analyses more clearly disentangle the contributions of each novel mechanism from those of the underlying MoE architecture and model capacity, and demonstrate that the added complexity of our method is both justified and controlled.

---

> > > ### Author Response · Authors · 2025-11-24
> > >
> > > **Question:** What are the theoretical guarantees or formal analyses regarding how the annealing schedule in CTA impacts expert load balancing or the stability/convergence of the MoE router?
> > >
> > > **Answer:** Thank you for the question. We have updated the theoretical analysis of the early-collapse problem of the router and how CTA addresses it in Appendix B.
> > >
> > > Specifically, we show that the gradient of the scores of **active** experts is expert-specific, as it depends on the comparison between the expert output and the MoE output. In contrast, the scores of **inactive** experts receive gradients that are not related to this comparison. As a result, the router does not obtain accurate gradients that reflect each inactive expert’s output.
> > >
> > > We further analyze the gradients of the scores of active experts and find that they are aligned with directions that reduce the training loss. Thus, among active experts, the score gradients tend to decrease the loss, whereas for inactive experts the score gradients may point to directions that either increase or decrease the loss.
> > >
> > > This suggests that, since the router’s score gradients for active experts are informative for improving performance, it is beneficial to first activate all experts, allow the router to discover better experts, and then gradually reduce the number of active experts. CTA follows this principle: it initially activates all experts more uniformly and then progressively narrows down the active set.
> > >
> > > In addition, we visualize the expert activation frequency during training. Without CTA, the activation frequencies change very little after 10–20 epochs and differ substantially across different random seeds. With CTA, although the activation frequencies are quite different at the beginning of training, they become much more consistent later on.
> > >
> > > **Question:** Beyond empirical performance, what is the formal algorithmic advantage of PER over a standard token-wise or layer-wise routing scheme in the context of dense visual representations?
> > >
> > > **Answer:** Patchwise Expert Routing (PER) assigns experts at the level of individual patch tokens, and applies the same principle in the distillation stage via teacher-specific routers. Concretely, the image is first divided into patches, a standard procedure in Vision Transformer–based encoders, and PER then selects the most suitable expert for each patch token.
> > >
> > > Teacher Routing builds on this by leveraging pre-trained teacher-specific routers. During distillation, we train separate routers for different teachers (e.g., a DINOv2 router, a ViT router, and a CLIP router). At inference, we can select among these teacher routers and use them to guide expert selection for each patch. Teacher Routing thus consists of Frame-wise Teacher Routing (FTR) and Layer-wise Teacher Routing (LTR). Since we have a distinct router for each MoE layer, we either (i) select one teacher for all MoE layers (FTR) or (ii) select potentially different teachers independently for each MoE layer (LTR).
> > >
> > > The key algorithmic advantage of PER lies in the size of its search space over expert assignments. PER can explore *all* possible expert combinations across patches and layers, whereas FTR and LTR are restricted to much smaller subsets. For example, suppose each image has 196 patches, we choose 2 experts out of 6 for each patch, there are 3 MoE layers, and we have 3 teacher routers. In this setting:
> > >
> > > - PER can, in principle, choose from
> > >   $(\binom{6}{2})^{196 \times 3} = \left(\frac{6 \times 5}{2}\right)^{588}$
> > >   distinct expert assignment configurations.
> > >
> > > - FTR has only 3 possible configurations, corresponding to choosing a single teacher (DINOv2, ViT, or CLIP) for all MoE layers.
> > >
> > > - LTR has $3^3 = 27$ possible configurations, since it chooses one teacher per layer independently.
> > >
> > > Thus, analytically, PER enjoys a dramatically larger combinatorial search space over expert assignments. This richer search space allows PER to approximate or discover more optimal combinations of experts across patches and layers, providing a formal algorithmic advantage over both standard token-wise and layer-wise teacher routing schemes in dense visual representation learning.

---

> > > > ### Author Response · Authors · 2025-11-24
> > > >
> > > > **Question:** The distillation loss uses mutual-information regularization. Please elaborate on the theoretical motivation for this specific regularization in this context.
> > > >
> > > > **Answer:** Thank you for the question. The main idea is that we want our router to reflect task semantics in how it assigns tokens to experts. To achieve this, we regularize the router using the mutual information between the teacher identity and the selected expert.
> > > >
> > > > Formally, let $I$ denote the teacher (i.e., which VFM is providing supervision) and $E$ denote the selected expert. We consider the mutual information $I(I; E) = H(I) - H(I \mid E)$. In our setting, we assume each VFM is equally important and is used with a roughly uniform prior, so $H(I)$ is (approximately) fixed. Under this assumption, maximizing $I(I;E)$ is equivalent to minimizing the conditional entropy $H(I \mid E)$.
> > > >
> > > > Minimizing $H(I \mid E)$ means that, once an expert $E$ is chosen and a token $x$ is given, the teacher identity $I$ is predictable (i.e., the uncertainty about which teacher is being used is small). Intuitively, this encourages each expert to specialize in and preserve the distinctive knowledge associated with a particular VFM, thereby reducing interference across VFMs for a given input token $x$.
> > > >
> > > > From an intuitive standpoint, we have two main goals. First, we want to preserve fine-grained vision representations from different VFMs. However, different VFMs produce embeddings in different feature spaces. If we only distill multiple VFMs using a token-balance loss [1], we merely equalize the token load across experts. This is **task-agnostic**: it does not align experts with any particular VFM or task semantics. As a result, the potentially conflicting supervision from different VFMs is mixed within the same experts during training, leading to compromised representations that do not faithfully preserve any single VFM.
> > > >
> > > > Second, mutual-information regularization provides more flexibility for downstream robot tasks. Because we use teacher-specific routers, each expert (and its associated router) can specialize in one VFM. This has two advantages: (i) we can reconstruct the vision representation of a specific VFM via its teacher-specific router (the three student representations in the top-right corner of Figure 2 illustrate that different teacher-specific routers produce distinct reconstructed vision representations), and then evaluate how each reconstructed representation performs on downstream tasks; and (ii) we can implement **Teacher Routing**, where we train an additional network to select among different teacher-specific routers at test time, effectively choosing between different VFM representations depending on the input or task.
> > > >
> > > > These capabilities, that recovering teacher-specific representations and enabling Teacher Routing, cannot be achieved with a pure token-balance loss. Related quantitative results supporting this design choice are shown in Table 3.
> > > >
> > > > [1] Shazeer, Noam, et al. "Outrageously large neural networks: The sparsely-gated mixture-of-experts layer." arXiv preprint arXiv:1701.06538 (2017).

---

### Official Review · Reviewer_We1f · 2025-10-29

**Soundness:** 2
**Presentation:** 3
**Contribution:** 3
**Rating:** 6
**Confidence:** 2

**Summary:**

The paper introduces VER (Vision Expert Transformer), a novel architecture for robot learning. It addresses the limited generalization of single Vision Foundation Models and the inflexibility of distilling multiple VFMs into a static representation. VER's core contribution is a two-stage framework: 1. Pretraining: Knowledge from multiple VFMs is distilled into a "Vision Expert Library" (VEL) based on a Mixture-of-Experts (MoE) architecture. 2. Fine-tuning: The pretrained experts are frozen, and only a lightweight "Robot Router" is trained to dynamically select task-relevant experts. To enhance expert selection, the authors propose "Patchwise Expert Routing" (PER) and "Curriculum Top-K Annealing" (CTA). CTA starts training with all experts active and gradually anneals the number of active experts to prevent premature convergence. Experiments show that VER achieves state-of-the-art performance.

**Strengths:**

- This work addresses a practical and important problem: efficiently leveraging multiple powerful VFMs for robotics. The ability to adapt to new tasks by fine-tuning only <0.4% of parameters is highly valuable for resource-constrained robotic systems.
- The work creatively combines multi-teacher VFM distillation with an MoE architecture.
- The paper is well-organized. The experimental evaluation is thorough and robust.

**Weaknesses:**

I'm sorry, but I'm not very familiar with this field. Please see the Questions section for details.

**Questions:**

1. The CTA schedule depends on the hyperparameter $S$. How was $S$ determined in your experiments? Is the method sensitive to this choice? I may have missed the relevant ablation studies. Additionally, are there any empirical conclusions that help in selecting this hyperparameter?
2. Could you explain the main differences between VER and the baseline Theia? Table 7 shows that Theia does not use the MoE architecture, while VER has 135% more parameters than Theia. Would it be feasible to replace Theia's network architecture with the MoE architecture for comparison?
3. Could you please elaborate on the training procedure for the "Train-from-Scratch" (TFS) experts in Table 5? Specifically, for the (#DFM=6, #TFS=1) setting, how is this 7th expert initialized and trained? Additionally, would fine-tuning all parameters across 6 DFMs yield better performance? Or introduce a LoRA module for fine-tuning?

---

> ### Author Response · Authors · 2025-11-24
>
> Thank you for highlighting the practical importance of efficiently leveraging multiple VFMs for robotics, the parameter-efficient adaptation, and the creative combination of multi-teacher distillation with an MoE architecture, as well as the thorough and well-organized experimental evaluation. The following are our responses to your questions.
>
> **Question:** The CTA schedule depends on the hyperparameter $S$. How was $S$ determined in your experiments? Is the method sensitive to this choice? I may have missed the relevant ablation studies. Additionally, are there any empirical conclusions that help in selecting this hyperparameter?
>
> **Answer:** Thank you for the question. We have updated the ablation study on $S$ and expanded the theoretical analysis of the early-collapse problem of the router, as well as how CTA mitigates this issue, in Appendix B. Overall, increasing $S$ enables the robot router to learn more effective expert selection, as illustrated in the following table. However, a larger $S$ also activates more experts during training, which leads to higher GPU memory usage and computational cost. Therefore, an appropriate choice of $S$ allows us to balance performance and computational efficiency.
>
> **Table:** Ablation study on $S$ in CTA. Success rate (%) is reported.
>
> | Task  | $S = 0$        | $S = 40$       | $S = 60$             | $S = 80$             |
> |-------|--------------|--------------|---------------------|---------------------|
> | pen-v0 | $77.3 ± 8.3$   | $78.7 ± 2.3$   | $\mathbf{81.3 ± 2.3}$      | $\mathbf{81.3 ± 2.3}$      |
>
>
> **Question:** Could you explain the main differences between VER and the baseline Theia? Table 7 shows that Theia does not use the MoE architecture, while VER has 135% more parameters than Theia. Would it be feasible to replace Theia's network architecture with the MoE architecture for comparison?
>
> **Answer:** Thank you for the question. The main differences between VER and the baseline Theia are as follows: (i) we replace the last three transformer layers with a Mixture-of-Experts (MoE) structure; (ii) during robot policy learning, we train a lightweight robot router to select task-relevant visual experts, whereas Theia provides only a unified visual representation; and (iii) we introduce CTA to further enhance performance on downstream robot tasks.
>
> VER has 135% more parameters than Theia primarily due to the MoE structure. However, the number of *active* parameters during inference is similar, so the computational cost at test time remains comparable. Moreover, we show that VER-Tiny, which has significantly fewer parameters, achieves performance comparable to Theia-Small and Theia-Base in Figure 9.
>
> Our VER variant without CTA can be viewed as a version of Theia whose backbone has been replaced with an MoE-based architecture. The following table presents the comparison results. We observe that VER already outperforms Theia due to the MoE structure, and incorporating CTA yields further performance gains.
>
> **Table:** Ablation study on each component of VER. Success rate (%) is reported.
>
> |                     | **Theia**         | **VER**              | **VER+CTA**             |
> |---------------------|-------------------|----------------------|-------------------------|
> | pen-v0       | $74.0 \pm 2.8$    | $78.0 \pm 6.3$       | **$80.8 \pm 5.3$**      |
> | relocate-v0  | $46.0 \pm 2.8$    | $47.6 \pm 5.1$       | **$56.4 \pm 6.9$**      |
>
>
> **Question:** Could you please elaborate on the training procedure for the "Train-from-Scratch" (TFS) experts in Table 5? Specifically, for the (#DFM=6, #TFS=1) setting, how is this 7th expert initialized and trained?
>
> **Answer:** Thank you for the question. The expert parameters are denoted as $W_1 \in \mathbb{R}^{L \times d_{\text{in}} \times d_{\text{out}}}$, where $L$ is the total number of pre-trained experts, $d_{\text{in}}$ is the input dimension, and $d_{\text{out}}$ is the output dimension. When we introduce a new TFS expert (e.g., in the (#DFM=6, #TFS=1) setting), we add an additional parameter tensor $W_2 \in \mathbb{R}^{1 \times d_{\text{in}} \times d_{\text{out}}}$, which is randomly initialized. During training, $W_1$ is kept frozen, and only $W_2$ is updated. Thus, the 7th expert is learned from scratch on the target task, while the original six experts remain fixed.

---

> > ### Comment · Reviewer_We1f · 2025-11-27
> >
> > Thank you for your responses. I am satisfied with the rebuttal, as the additional information and clarifications have strengthened the paper's arguments. I am inclined to accept the paper; however, given that I am not an expert in this specific domain, I defer to the assessments of the other reviewers and will not be raising my score. I maintain my original rating of 6.

---

> > > ### Author Response · Authors · 2025-11-27
> > >
> > > Thank you so much for your follow-up and for carefully considering our rebuttal. We appreciate your constructive feedback, which has helped us strengthen and clarify the paper, and we are grateful for your efforts in reviewing work slightly outside your main area of expertise. We will incorporate the improvements discussed in our response into the final version. Thanks again!

---

### Official Review · Reviewer_KMyG · 2025-11-01

**Soundness:** 3
**Presentation:** 3
**Contribution:** 2
**Rating:** 6
**Confidence:** 3

**Summary:**

This paper investigates how to distill a vision encoder from existing vision foundation models for downstream robotic tasks. Unlike existing works that distill a monolithic encoder, this paper utilizes MoE to enable a more diverse, flexible and adaptive feature extraction process. Experimental results show that the proposed method outperforms baseline methods, especially Theia, on a broad range of tasks.

**Strengths:**

1. The paper is clearly motivated and presented.
2. The proposed method is evaluated on a broad range of tasks and shows clear advantage compared to baseline methods.
3. The ablation analysis and visualization are thorough and illustrative to demonstrate how the visual representation differs across different methods and the importance of different design choices.

**Weaknesses:**

My main concern about this paper is that its novely seems not very significant, more like just replacing existing distillation models for visual encoder in robotic models with MoE. The empirical gain is significant, but I think if we care about absolute performance, it may be better to compare with a broader range of methods for robotic control, such as VLAs that are fine-tuned on the same datasets. For other points, see my questions below.

**Questions:**

1. It would be good to add a qualitative analysis on the failure reasons of different methods.
2. Do you fine-tune the vision backbone of the baselines when learning the policy? If yes, do you fine-tune all their parameters? Or use some parameter-efficient methods like LoRA?
3. The problem setting and training pipeline are different, but how does your method compare with VLA methods that first pretrain on large-scale robotic data, then fine-tune on specific downstream tasks? Can your method be extended to replace the visual encoders used in VLA? If yes, does it perform better or not?
4. What is the size of your model at the three different parameter scale? Do you use a smaller MLP dimension in MoE so that the total parameter number is similar to that of Theia? As from Figure 9 it seems that your method has a smilar number of total parameters as Theia, but from Table 4 it seems that your method has a similar number of activated parameters (K=2) as Theia.
5. What will happen if you put more layers in your ViT as MoE layers?
6. As Theia is an important baseline you compare to, I think it's better to give a more detailed introduction on how Theia works for consistency of the paper, so we better know how your method differs from it.

---

> ### Author Response · Authors · 2025-11-24
>
> Thank you for recognizing that our method is well motivated, with strong experimental results and thorough visualizations. Since a main focus is the robot routing selection mechanism, we have added further analysis of the early collapse problem and additional visualizations in Appendix B and Appendix E. We sincerely appreciate your constructive comments, and the following are our detailed responses.
>
> **Weakness:** The empirical gain is significant, but the novelty seems not very significant. The proposed method is just replacing existing distillation models for the visual encoder in robotic models with MoE.
>
> **Response:** We appreciate your recognition of the empirical gains brought by our method. While we do introduce an MoE structure into the transformer-based visual encoder, our contribution goes beyond a simple architectural substitution by proposing new mechanisms in the distillation and robot policy learning pipelines, and by providing theoretical and empirical analyses focused on the *robot router* and its role in selecting task-relevant visual representations.
>
> 1. Router fine-tuning as a task-aware planning module
>
>    In our framework, we fine-tune the *robot router* to select experts specifically for downstream robot tasks. This is not merely a drop-in MoE replacement: the router is explicitly optimized to act as a *planning module* that chooses suitable intermediate visual representations conditioned on the task. Unlike conventional vision encoders, which provide a single, fixed representation, our robot router adaptively routes tokens to experts so that the policy receives task-specialized visual features. This router-centric adaptation is unique among existing visual encoders used in robotic systems.
>
> 2. Addressing router collapse via CTA and theoretical analysis
>
>    In Appendix B, we identify and theoretically analyze a *router collapse* problem specific to robot policy learning with MoE. Training the robot router is non-trivial because the policy head is randomly initialized and typically much larger than the router. In this setting, if the router prematurely converges, it tends to overuse a small subset of experts and fails to explore inactive experts effectively. *Theoretical analysis* in Appendix B shows that for inactive experts, the gradient of the corresponding router score does not reliably reflect the quality of those experts. As a result, once the router has converged, it becomes difficult to re-activate or explore inactive experts, especially when the policy head has not yet converged. Motivated by this analysis, we propose *CTA* as a principled strategy to anneal the top-k expert selection during training. CTA encourages exploration of multiple experts in the early stages and gradually sharpens the routing as training progresses, which both stabilizes training and improves final performance. This addresses a challenge that naive MoE structures face in robot learning and constitutes a key methodological contribution of our work.
>
> 3. Mutual-information-based objective instead of token load-balancing loss
>
>    Our training objective for the MoE visual encoder is also different from standard MoE practices in large language models. Instead of using the widely adopted *token load-balancing loss* [1], we employ a *mutual information (MI) loss* to train the MoE. This design choice is motivated by the need to *preserve fine-grained vision features* from the underlying VFM (Vision Foundation Model). The MI-based objective encourages experts to retain task-relevant information while avoiding unnecessary homogenization of representations. This contrasts with typical MoE LLM training, where load-balancing is primarily used to ensure uniform expert utilization and computational efficiency, rather than preserving nuanced visual details crucial for robot control.
>
> 4. In-depth analysis of task-relevant vision representations after routing
>
>     Beyond performance metrics, we conduct an in-depth analysis of how vision representations change before and after robot routing (see Appendix E.3). We show that, after routing, the representation predominantly preserves information in *task-relevant regions* while suppressing task-irrelevant content. For instance, in the rightmost image of the first row of Figure 7, the high-information region concentrates on the middle-right area, corresponding to the pose of the target pen. In the pen task, the robot must rotate the pen to align with this target pose. Our analysis reveals that the robot router learns to preserve precisely this task-relevant information and discard background details, which appear as dark, low-information regions.  We argue that these insights into how the router modulates visual representations are important for understanding and advancing robot learning.

---

> ### Author Response · Authors · 2025-11-24
>
> **Question:** Do you fine-tune the vision backbone of the baselines when learning the policy?
>
> **Answer:** No, we don't. We freeze all the parameters of our baselines when learning the policy.
>
> **Question:** What is the size of your model at the three different parameter scale? Do you use a smaller MLP dimension in MoE so that the total parameter number is similar to that of Theia? As from Figure 9 it seems that your method has a smilar number of total parameters as Theia, but from Table 4 it seems that your method has a similar number of activated parameters ($K=2$) as Theia.
>
> **Answer:** Thank you for the questions. The model sizes at the three scales are:
>
> - VER-Tiny (VER-T): 7.0M total / 5.3M active parameters
> - VER-Small (VER-S): 27.7M total / 20.8M active parameters
> - VER-Base (VER-B): 110.1M total / 82.2M active parameters
>
> Here, “total” denotes the parameters including all experts, while “active” denotes the parameters used at inference given top-K routing.
>
> Since we activate $K = 2$ experts at inference, we divide the MLP dimension of each expert by $2$ so that the number of active parameters is similar to that of Theia. In this way, we keep the effective inference-time capacity comparable while benefiting from the MoE structure.
>
> Figure 9 reports the trainable parameters versus success rate in the policy learning stage. For Theia, only the policy head is trainable. For VER, the trainable parameters include both the lightweight robot router and the policy head. Because the robot router is very small, the *overall trainable parameter counts are similar*, which explains the curves in Figure 9.
>
> The following table summarizes the trainable parameters for policy learning, showing robot router is lightweight:
>
> **Table:** Trainable parameters for policy learning with VER-T, VER-S, and VER-B.
>
> |                                      | VER-T| VER-S | VER-B |
> |--------------------------------------|-----------|-----------|-----------|
> | Total trainable parameters (M)   | 3.81      | 6.25      | 11.31     |
> | Robot router parameters (M)      | 0.03      | 0.11      | 0.45      |
>
> **Question:** As Theia is an important baseline you compare to, I think it's better to give a more detailed introduction on how Theia works for consistency of the paper, so we better know how your method differs from it.
>
> **Answer:** Thank you for your suggestion. We will update the manuscript to include a more detailed description of Theia for clearer and more consistent comparison. Here, we briefly highlight the key differences between Theia and our method (VER):
>
> (i) We introduce a *Mixture-of-Experts (MoE)* structure into the visual encoder to create a vision expert library, whereas Theia uses a single, shared encoder and output a unified representation.
>
> (ii) During robot policy learning, we train a *lightweight robot router* to select task-relevant visual experts, while Theia provides a single unified visual representation. As a result, Theia produces *task-agnostic* visual features, whereas VER provides *task-specific* visual representations (see Figure 8).
>
> (iii) We introduce *CTA* to further improve performance on downstream robot tasks by stabilizing and enhancing expert selection during training (see Appendix B).
>
> [1] Shazeer, Noam, et al. "Outrageously large neural networks: The sparsely-gated mixture-of-experts layer." arXiv preprint arXiv:1701.06538 (2017).

---

> ### Author Response · Authors · 2025-11-26
>
> **Question:** The problem setting and training pipeline are different, but how does your method compare with VLA methods that first pretrain on large-scale robotic data, then fine-tune on specific downstream tasks?
>
> **Answer:** We compare our VER + flow-matching policy head against two baselines: (1) Theia + flow-matching policy and (2) the fine-tuned VLA, GR00T N1.5 [1]. We evaluate performance on three downstream manipulation tasks: *place cross into bin*, *place cube into cup*, and *place cylinder onto plate*. For each task, we use 500 demonstrations. Following the recommended fine-tuning parameters, we fine-tune GR00T N1.5 for 20k steps with a batch size of 16. The results are summarized in the table below:
>
> | Model      | place cross into bin | place cube into cup | place cylinder onto plate |
> |-----------|-----------------------|---------------------|---------------------------|
> | Theia     | $0.65$                | $0.50$              | $0.70$                    |
> | GR00T N1.5 | $0.75$                | $0.73$              | $0.70$                    |
> | VER       | $\mathbf{0.95}$       | $\mathbf{0.75}$     | $\mathbf{0.85}$           |
>
> We observe that VER outperforms Theia and Gr00tN1.5 across all three tasks, indicating that our method is more effective in this experimental setup.
>
> [1] NVIDIA et al., “GR00T N1: An Open Foundation Model for Generalist Humanoid Robots,” arXiv preprint arXiv:2503.14734, March 2025.

---

> ### Author Response · Authors · 2025-11-27
>
> **Question:** What will happen if you put more layers in your ViT as MoE layers?
>
> **Answer:** Thank you for the question. Our VER architecture consists of two components: a Base Vision Transformer (BVT) with $M$ standard transformer layers and a Vision Expert Library (VEL) with $N$ MoE layers. Increasing the number of MoE layers corresponds to increasing $N$.
>
> We use 25% of the ImageNet-1K dataset and pre-train VER-Tiny under different $(M, N)$ configurations for 50 epochs. We then evaluate on the pen-v0 robot task by freezing all VER parameters and training only the robot router and policy head over five random seeds. We also report the average cosine distillation loss to DINOv2, ViT, and CLIP.
>
> Table: Ablation study on different $M$ and $N$
> | $M$ | $N$ | Distill cosine loss | Success Rate (%)        |
> |-----|-----|---------------------|-------------------------|
> | $7$ | $3$ | $0.561$             | $50.4 \pm 12.5$         |
> | $9$ | $3$ | $0.551$             | $69.6 \pm 4.8$          |
> | $9$ | $5$ | $0.546$             | $48.8 \pm 12.2$         |
>
> In this setting, increasing the number of BVT layers from $M = 7$ to $M = 9$ (with $N = 3$ fixed) improves the distillation loss and leads to higher success rates on the downstream task. When we further increase the number of MoE layers from $N = 3$ to $N = 5$ (with $M = 9$ fixed), the distillation loss continues to improve, but the downstream success rate decreases. These preliminary results suggest that, under our current training setup, using more MoE layers does not necessarily improve downstream performance, and a moderate number of MoE layers (e.g., $N = 3$) is preferable. One possible reason is that deeper MoE stacks make the robot routing problem more challenging, as the space of possible expert combinations grows exponentially with the number of MoE layers (e.g., if a single layer has $a$ possible combinations, five stacked layers yield $a^5$ combinations). Moreover, more layers mean more parameters, which can improve distillation quality but also increase GPU memory usage and computation cost. Thus, a suitable number of layers is preferred to balance performance and efficiency.

---

### Official Review · Reviewer_dKGG · 2025-11-01

**Soundness:** 4
**Presentation:** 4
**Contribution:** 3
**Rating:** 8
**Confidence:** 4

**Summary:**

This paper addresses the limitation that single vision foundation models (VFMs) cannot meet the diverse requirements of robotic tasks, and existing multi-model distillation methods dilute model-specific capabilities while lacking task adaptability. The proposed VER framework distills multiple VFMs into a Vision Expert Library and employs a dynamic routing mechanism to select the most relevant visual experts for different robotic tasks. The first core innovation is Patchwise Expert Routing combined with Curriculum Top-K Annealing (PER+CTA), which achieves fine-grained patch-level expert selection and prevents early training collapse through curriculum-based K-value annealing. The second core innovation involves fine-tuning only a lightweight Router (<0.4% parameters) while freezing all pretrained experts, enabling parameter-efficient task adaptation and flexible integration of trainable experts to incorporate robot-domain knowledge. Experimental validation across 17 robotic tasks achieves state-of-the-art performance (74.7% average success rate) and outperforms existing methods across multiple policy heads (ViLT, Flow-Matching, Diffusion). A key finding is that dynamic routing suppresses large-norm outliers in task-irrelevant regions and concentrates attention on task-critical areas (such as target object poses), generating more compact and discriminative visual features.

**Strengths:**

Originality and Problem Formulation: The paper presents a novel paradigm shift from static unified representations to dynamic expert selection for robot learning, introducing the innovative combination of vision foundation model distillation with Mixture-of-Experts (MoE) architecture and the creative Curriculum Top-K Annealing mechanism that addresses the early collapse problem in lightweight router training.

Quality and Experimental Rigor: The work demonstrates exceptional experimental rigor with comprehensive evaluation across 17 diverse robotic tasks spanning multiple benchmarks (Franka Kitchen, Meta-World, Adroit, LIBERO), different policy heads (ViLT, Flow-Matching, Diffusion), and real-world experiments, complemented by thorough ablation studies and insightful patch-level feature analysis (entropy, mutual information) that reveal the underlying mechanism of how dynamic routing concentrates attention on task-critical regions.

Clarity and Significance: The paper is well-structured with clear motivation and methodology, providing strong evidence that VER achieves state-of-the-art performance (74.7% average success rate) while maintaining computational efficiency (only <0.4% trainable parameters during downstream tasks, identical inference time to baselines), and the framework's extensibility—enabling flexible Top-K adjustment for compute scaling and seamless integration of task-specific trainable experts—offers significant practical value for scalable robot learning systems.


In addition, the experimental results in Figures 6 and 8 (1) are one of the reasons I gave you a high score.

**Weaknesses:**

Arbitrary choices lacking design principles: The paper fixes L=6 experts and K=2 active experts without providing any ablation studies to justify these hyperparameter choices (e.g., why not L=3,K=1 or L=9,K=3), and the "coincidence" that L=6 is exactly twice the number of three teachers suggests a potential pre-assigned expert-teacher correspondence, contradicting the paper's claim of "dynamic allocation."
Missing critical architecture search: The paper fails to explore the joint impact of L and K (e.g., performance comparison of L=3,K=1 vs L=6,K=2 vs L=12,K=4 under the same computational budget), making it impossible to determine whether the performance gains stem from the inherent advantages of the MoE architecture or merely from accidentally selecting a configuration that works for this specific task set, severely undermining the method's generalizability and reproducibility. But these are not big issues.

**Questions:**

Can you explore Inference Speed and Efficiency?

I recommend acceptance, but cannot confirm yet whether a higher score can be given.

---

> ### Author Response · Authors · 2025-11-24
>
> Thank you very much for recognizing the novelty of our VER framework, especially the idea of turning vision representations into a library rather than a single unified model, and enabling dynamic routing to select suitable visual experts. We also sincerely appreciate that you highlighted the rigorous evaluation, the practical value of our approach, and the clarity of the writing. In the revised version, we provide more rigorous theoretical analysis of the router and the Curriculum Top-K Annealing (CTA) in Appendix B, as well as additional visualizations and analyses of the vision representations in Appendix E. The following are our detailed responses to your comments.
>
> **Weakness:** Arbitrary choices on number of active experts and total experts lacking design principles.
>
> **Response:** Thank you for this thoughtful comment and for explicitly noting that these are not major issues. We agree that a more systematic exploration of $L$, $K$ would be valuable, and we clarify our design choices and scope below.
>
> 1. Design rationale for $L = 6$ and $K = 2$.
>    Our primary goal in this work is to study how MoE-based visual representations can benefit downstream *robot policy learning*, and how the robot router’s selection of visual representations improves robot performance and provides insight into suitable visual features for robot tasks. We chose $L = 6$ and $K = 2$ to balance the following considerations:
>    - Allow sufficient *expert diversity* and *expert-selection diversity*, so that the router can discover non-trivial specializations beyond a one-to-one teacher mapping and exploit a richer set of expert combinations.
>    - Avoid additional computation cost when scaling up $L$.
>
> 2. On the “coincidence” of $L = 6$ and three teachers.
>    Although $L = 6$ is twice the number of teachers, there is *no pre-assigned expert–teacher correspondence* in our implementation. The router is trained purely via the distillation loss and the mutual information loss. The allocation is *dynamic*, and experts are not hard-bound to specific teachers. As shown in Figure 4, the ViT teacher tends to activate fewer experts, whereas CLIP and DINOv2 activate more experts. This reflects the varying difficulty of mimicking different VFMs (Table 8 also shows that ViT has a lower distillation loss) and empirically supports that the router performs dynamic allocation rather than fixed teacher–expert matching.
>
> 3. Scope and limitations of architecture search.
>    We acknowledge that we do not include a full grid search over $L$, $K$ (e.g., $L = 3$, $K = 1$; $L = 6$, $K = 2$; $L = 12$, $K = 4$) under matched compute. The main reason is *computational cost*: each configuration requires full distillation and evaluation across multiple robot tasks, which is prohibitive under our resources. Since our main goal is to test whether *robot routing*—i.e., selecting suitable visual representations—can benefit robot tasks and to understand the associated challenges, we fixed $L$, $K$ and focused our experimental budget on:
>    - Analyzing *router behavior and collapse* and proposing CTA.
>    - Studying how the router shapes *task-relevant visual representations* and how this affects robot success rates.
>
>
> In addition, we provide theoretical analysis of the early collapse problem of the robot router in Appendix B, and we run more experiments to visualize and validate the router collapse phenomenon during policy learning, showing how CTA leads to more stable and consistent expert allocation for downstream robot tasks. In the following response, we have included the ablation study on  different MoE $K$ and $L$ configuration. Hope this response and following experiment results can address your concern.

---

> ### Author Response · Authors · 2025-11-24
>
> **Question:** Can you explore Inference Speed and Efficiency?
>
> **Answer:** Thank you for your question. With the diffusion head, the inference time of the diffusion policy is approximately $0.105 s$ on an RTX 4090 for both VER and Theia. To more clearly illustrate the latency behavior of our MoE design, we further measure the latency of different $K$, $L$ configurations on *VER-Tiny* using an *untrained model* with *batch size = 1*. We record the inference time of BVT (Base Vision Transformer consisting of 9 vanilla transformer layers) and a 3-layer MoE. For the MoE, we record the inference time of the router and the experts separately. Note that in the MoE architecture there are also attention layers, which contribute to the overall inference time.
>
> **Table:** Latency (**ms**) comparison for different $K$, $L$ configurations.
>
> | $K$ | $L$  | BVT (9 Layers) | MoE (3 Layers) | Router | Expert    |
> |---|----|----------------|----------------|--------|--------|
> | $1$ | $3$  | $3.2430$         | $1.4491$         | $0.5620$ | $0.5464$ |
> | $2$ | $4$  | $3.2650$         | $1.4873$         | $0.5526$ | $0.6021$ |
> | $2$ | $6$  | $3.2683$         | $1.6227$         | $0.5855$ | $0.6939$ |
> | $3$ | $9$  | $3.2759$         | $1.7693$         | $0.5864$ | $0.8524$ |
> | $4$ | $12$ | $3.2836$         | $1.9383$         | $0.5847$ | $1.0089$ |
>
> From the table, we observe that as $L$ increases, the *MoE latency* also increases. As $L$ increases, the expert’s latency increases; as $K$ increases, the routing latency increases. Overall, our choice of ($K$ = 2, $L$ = 6) represents a practical balance between *model capacity* and *latency cost* under our computational constraints.

---

> ### Author Response · Authors · 2025-11-26
> **Additional experiments on MoE configuration (K and L)**
>
> **Weakness:** Arbitrary choices of the number of active experts and total experts lacking design principles.
>
> **Response:** Thank you for your comments. To address this concern, we extended our experiments to systematically study different $(K, L)$ configurations. Specifically, we use 25% of the ImageNet-1K dataset and pre-train our VER-Tiny model under varying numbers of active experts $K$ and total experts $L$. Following the reviewer’s suggestion, we include configurations $(K, L) = (1, 3), (2, 6), (4, 12)$, as well as an intermediate setting $(2, 4)$.
>
> To ensure a fair comparison under an approximately constant computational budget, we divide each expert MLP’s hidden dimension by $K$, so that the number of active parameters per forward pass is kept roughly constant across different $(K, L)$ settings. We pre-train VER for 50 epochs. For evaluation, we choose the pen-v0 robot task, freeze all parameters in VER, and train only the robot router and policy head over five random seeds.
>
> The results are summarized below:
>
> | $K$ | $L$  | Success Rate (%)        |
> |-----|------|-------------------------|
> | $1$ | $3$  | $44.0 \pm 11.9$         |
> | $2$ | $4$  | $48.8 \pm 5.3$          |
> | $2$ | $6$  | $69.6 \pm 4.8$          |
> | $4$ | $12$ | $66.4 \pm 4.1$          |
>
> We observe that performance improves as we move from $(K, L) = (1, 3)$ to $(2, 6)$, and then slightly degrades at $(4, 12)$. This suggests that a moderate number of active experts and total experts (here, $K = 2$, $L = 6$) achieves the best performance under a fixed active-parameter budget. Too few experts reduce expert diversity and lead to more homogeneous representations, while too many experts can slightly hurt performance, likely due to increased routing and optimization complexity. These findings provide a more principled justification for our choice of $K = 2$, $L = 6$ as a good trade-off between expert diversity and optimization difficulty, rather than an arbitrary configuration. Finally, as discussed in our previous response, increasing $K$ and $L$ also increases latency, highlighting a trade-off between performance and inference efficiency.

---

### Author Response · Authors · 2025-12-03

Dear Reviewers,

We would like to express our sincere gratitude to all reviewers for the time and effort invested in evaluating our paper and engaging with us to improve its quality. Although this unexpected situation has occurred, we, as the authors, deeply appreciate your thoughtful reviews and your contributions during the rebuttal process.

We are pleased that reviewers generally acknowledged the contributions of our work:

- **Idea.** Reviewers noted that the paper (i) introduces a novel shift from static unified representations to dynamic expert selection for robot learning and a creative Curriculum Top-K Annealing mechanism that mitigates early collapse [dkGG]; (ii) is clearly and convincingly motivated [KMyG]; and (iii) addresses an important practical problem and makes a substantial contribution to robot learning [We1f, b4k1].

- **Experiment.** Reviewers found the experiments thorough and robust, with strong performance demonstrated across diverse tasks and policy heads [dkGG, KMyG, We1f, b4k1].

- **Complete analysis.** Reviewers highlighted the analysis of vision features after dynamic selection and mutual information [dkGG], and considered the ablations and visualizations thorough and illustrative in revealing representation differences across methods and the impact of key design choices [KMyG].

- **Writing.** Reviewers described the paper as clear and significant [dkGG], clearly presented [KMyG], and well organized [We1f].

We also thank all reviewers for their insightful and constructive suggestions. Below we summarize the supporting experiments and analyses added in the rebuttal in response to your feedback:

- Theoretical analysis of early collapse of the robot router and the rationale of the CTA algorithm (Appendix B) [b4k1, KMyG]
- Theoretical analysis of mutual information loss (Appendix D.2) [b4k1, KMyG]
- Additional vision feature visualizations and mutual information analysis (Appendix E.3)
- Ablation on the number of active experts and the total number of experts [dkGG]
- Comparison of VER with Vision-Language-Action Models [KMyG]
- Ablation on the number of MoE layers and the depth of the base Vision Transformer [KMyG]
- Ablation on $S$, the annealing schedule parameter in Curriculum Top-K Annealing [We1f]

We hope these new additions help address your concerns and better position our work. We are very grateful for the reviewers’ time and feedback in improving the quality of our paper. Although we cannot receive further responses from you at this stage, we are continuing to refine our rebuttal and manuscript to address your comments as fully as possible. We highly value your praise, suggestions, and comments.

Thank you again for your time and efforts.

Best,
Authors

---

### Author Response · Authors · 2025-12-03

Dear ACs,

We would like to express our sincere gratitude to you for taking on the responsibility of handling our paper during this unexpected and challenging situation. Your efforts are crucial to maintaining ICLR’s standards of quality and fairness, and we deeply appreciate the additional workload this entails. To help with the assessment, we briefly summarize how the reviewers have viewed the paper and how we have strengthened it during the rebuttal phase.

Reviewers generally recognized key strengths of our work: (i) a novel shift from static unified representations to dynamic expert selection for robot learning, together with a Curriculum Top-K Annealing mechanism to mitigate early collapse [dkGG]; (ii) clear and convincing motivation [KMyG]; (iii) strong and robust empirical performance across diverse tasks and policy heads [dkGG, KMyG, We1f, b4k1]; (iv) comprehensive analyses of vision features under dynamic selection and mutual information [dkGG, KMyG]; and (v) clear, well-organized writing [dkGG, KMyG, We1f].

In response to their constructive feedback, we further reinforced the paper with additional theoretical analysis of early collapse and the rationale of the CTA algorithm (Appendix B) and mutual information loss (Appendix D.2) [b4k1, KMyG], extended visualizations and mutual-information-based feature analysis (Appendix E.3), ablations on the number of experts and model depth, studies of the annealing schedule parameter $S$, and new comparisons between VER and Vision-Language-Action models.


In addition, we summarize the main outcomes of the rebuttal phase with the reviewers:

- Reviewer [dkGG] indicated that our rebuttal addressed their main concerns and subsequently raised their score after we added the additional experiments.
- Reviewer [We1f] also expressed satisfaction with our rebuttal and the new results.
- Unfortunately, we have not yet received further responses from Reviewer [KMyG] and Reviewer [b4k1]. We hope that the additional mathematical analyses, which further support the rationale of our approach, together with the new experiments (including comparisons with VLAs), adequately address their primary concerns.

We are very grateful for the care, time, and professionalism you invest in this process. We sincerely appreciate your efforts to ensure a fair, thorough, and principled evaluation of our work.

Thank you again for your time and consideration.

Best,
Authors

---

### Meta-Review · Area_Chair_L9qS · 2026-01-07

**Summary:**

Reviewers generally found this paper to be well motivated, clearly written, and empirically strong, with extensive evaluation across 17 robotic tasks, multiple policy heads, and both simulated and real-world settings. The strongest reviewer (dKGG) highlighted the practical significance, robust experimental design, and the effectiveness of dynamic expert routing in focusing on task-relevant visual regions. While some reviewers raised concerns about limited conceptual novelty, framing the method as an MoE-based extension of existing distillation approaches, the paper introduces several non-trivial methodological contributions—notably patchwise expert routing, curriculum top-K annealing to mitigate router collapse, and a mutual-information-based distillation objective—supported by detailed empirical analysis and additional ablations in the rebuttal.

Overall, the work demonstrates that task-aware visual routing yields consistent and meaningful gains for robot learning, and the rebuttal strengthened the justification of design choices and clarified comparisons with strong baselines, including VLAs. It's worthy of publication for wider audience to be aware.

**Reviewer Concerns:**

Addressed:
- Hyperparameter choices and architectural design (K, L, CTA schedule, number of MoE layers) via additional ablations
- Router stability and early-collapse concerns through theoretical analysis and empirical validation of CTA
- Fairness and clarity of comparisons with Theia and VLA-style methods
- Model size, active vs. total parameters, and inference efficiency

Remaining:
- Novelty concerns remain for some reviewers, particularly regarding whether the core idea represents a paradigm shift versus a principled integration of MoE and distillation
- Theoretical analysis, while improved, remains mostly intuitive rather than offering formal guarantees

**Reviewer Scores:**

- Reviewer dKGG: Would likely increase or remain high
- Reviewer We1f: Would remain 6 (explicitly satisfied with rebuttal)
- Reviewer KMyG: Would likely remain 6
- Reviewer b4k1: Might increase slightly (2 → 3), concerns are primarily about novelty rather than correctness

---

### Decision · Program_Chairs · 2026-01-26

Accept (Poster)